# PIXIE: FAST AND GENERALIZABLE SUPERVISED LEARNING OF 3D PHYSICS FROM PIXELS

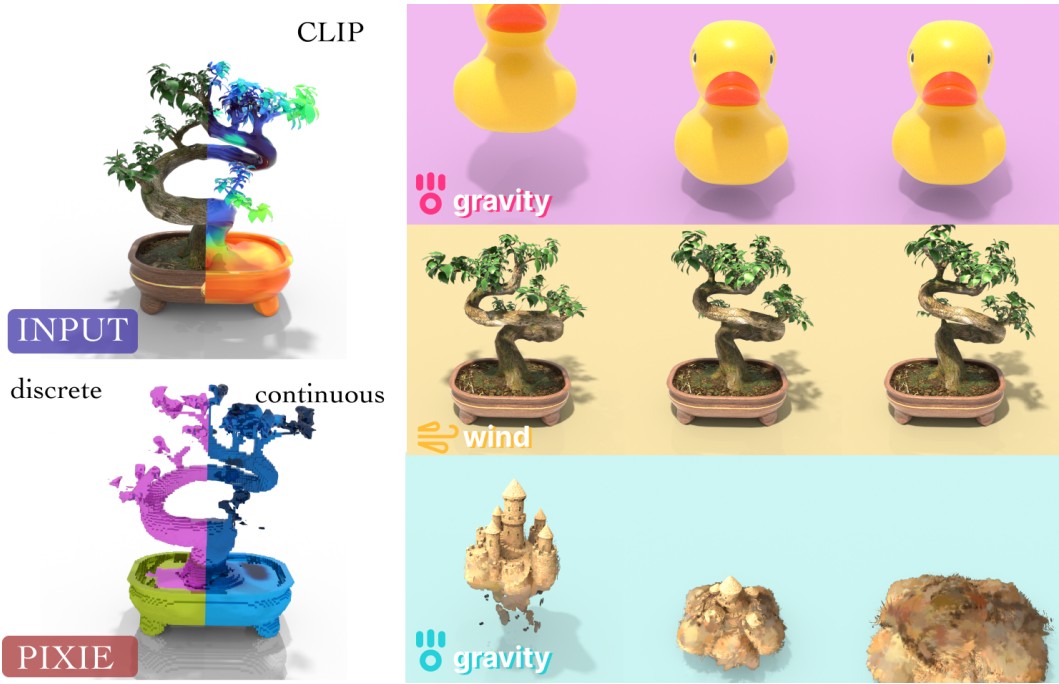

Figure 1: We introduce PIXIE, a novel method for learning simulatable physics of 3D scenes from visual features. Trained on a curated dataset of paired 3D objects and physical material annotations, PIXIE can predict both the discrete material types (e.g., rubber) and continuous values including Young's modulus, Poisson's ratio, and density for a variety of materials, including elastic, plastic, and granular. The predicted material parameters can then be coupled with a learned static 3D model such as Gaussian splats and a physics solver such as the Material Point Method (MPM) to produce realistic 3D simulation under physical forces such as gravity and wind.

## ABSTRACT

Inferring the physical properties of 3D scenes from visual information is a critical yet challenging task for creating interactive and realistic virtual worlds. While humans intuitively grasp material characteristics such as elasticity or stiffness, existing methods often rely on slow, per-scene optimization, limiting their generalizability and application. To address this problem, we introduce PIXIE, a novel method that trains a generalizable neural network to predict physical properties across multiple scenes from 3D visual features purely using supervised losses. Once trained, our feed-forward network can perform fast inference of plausible material fields, which coupled with a learned static scene representation like Gaussian Splatting enables realistic physics simulation under external forces. To facilitate this research, we also collected PIXIEVERSE, one of the largest known datasets of paired 3D assets and physic material annotations. Extensive evaluations demonstrate that PIXIE is about 1.46-4.39x better and orders of magnitude faster than test-time optimization methods. By leveraging pretrained visual features like CLIP, our method can also zero-shot generalize to real-world scenes despite only ever been trained on synthetic data. https://pixie-2026-12998.github.io/

## 1 INTRODUCTION

Advances in scene reconstruction with Neural Radiance Fields (Mildenhall et al., 2021) and Gaussian Splatting (Kerbl et al., 2023) have made it possible to recreate photorealistic 3D world from sparse camera views, with broad applications from immersive content creation to robotics and simulation. However, these approaches focus exclusively on visual appearance—capturing the geometry and colors of a scene while remaining blind to its underlying physical properties.

Yet the world is not merely a static collection of shapes and textures. Objects bend, fold, bounce, and deform according to their material composition and the forces acting upon them. Consequently, there has been a growing body of work that aims to integrate physics into 3D scene modeling (Pumarola et al., 2020; Ma et al., 2023; Li et al., 2023; Fischer et al., 2024; Feng et al., 2023; Xie et al., 2023; Qiu et al., 2024; Guo et al., 2024; Lin et al., 2025; Zhai et al., 2024; Chen et al., 2025b). Current approaches for acquiring the material properties of the scene generally fall into two categories, each with significant limitations. Some works such as (Xie et al., 2023; Guo et al., 2024) require users to manually specify material parameters for the entire scene based on domain knowledge. This manual approach is limited in its application as it places a heavy burden on the user and lacks fine-grained detail. Another line of work aims to automate the material discovery process via test-time optimization. Works including (Jatavallabhula et al., 2021; Li et al., 2023; Zhong et al., 2024; Huang et al., 2024; Lin et al., 2025; Zhang et al., 2024) leverage differentiable physics solvers, iteratively optimizing material fields by comparing simulated outcomes against ground-truth observations or realism scores from video generative models. However, predicting physical parameters for hundreds of thousands of particles from sparse signals (i.e., a single rendering or distillation scalar loss) is an extremely slow and difficult optimization process, often taking hours on a single scene. Furthermore, this heavy per-scene memorization does not generalize: for each new scene, the incredibly slow optimization has to be run from scratch again.

In this paper, we propose a new framework, PIXIE, which unifies geometry, appearance, and physics prediction via direct supervised learning. Our approach is inspired by how humans intuitively understand physics: when we see a tree swaying in the wind, we do not memorize the stiffness values for each specific coordinate $(x, y, z)$ – instead, we learn that objects with tree-like visual features behave in certain ways when forces are applied. This physical understanding from visual cues allows us to anticipate the motion of a different tree or even other vegetation like grass, in an entirely new context. Thus, our insight is to leverage rich 3D visual features such as those distilled from CLIP (Radford et al., 2021) to predict physical materials in a direct supervised and feed-forward way. Once trained, our model can associate visual patterns (e.g., "if it looks like vegetation") with physical behaviors (e.g., "it should have material properties similar to a tree"), enabling fast inference and generalization across scenes. To facilitate this research, we have curated and labeled PIXIEVERSE, a dataset of 1624 paired 3D objects and annotated materials spanning 10 semantic classes. We developed a multi-step and semi-automatic data labeling process, distilling pretrained models including Gemini (Team et al., 2023), CLIP (Radford et al., 2021), and human priors into the dataset. To our knowledge, this is the largest open-source dataset of paired 3D assets and physical material labels. Trained on PIXIEVERSE, our feed-forward network can predict material fields that are 1.46-4.39x better and orders of magnitude faster than test-time optimization methods. By leveraging pretrained visual features, PIXIE can also zero-shot generalize to real-world scenes despite only ever being trained on synthetic data.

Our contributions include:

1. **Novel Framework for 3D Physics Prediction**: We introduce PIXIE, a unified framework that predicts discrete material types and continuous physical parameters (Young's modulus, Poisson's ratio, density) directly from visual features using supervised learning.
2. **PIXIEVERSE Dataset**: We curate and release PIXIEVERSE, the largest open-source dataset of 3D objects with physical material annotations (1624 objects, 10 semantic classes).
3. **Fast and Generalizable Inference**: By leveraging pretrained visual features from CLIP and a feed-forward 3D U-Net, PIXIE performs inference orders of magnitude faster than prior test-time optimization approaches, achieving a 1.46-4.39x improvement in realism scores as evaluated by a state-of-the-art vision-language model.

4. **Zero-Shot Generalization to Real Scenes**: Despite being trained solely on synthetic data, PIXIE generalizes to real-world scenes, showing how visual feature distillation can effectively bridge the sim-to-real gap.

5. **Seamless Integration with MPM Solvers**: The predicted material fields can be directly coupled with Gaussian splatting models for realistic physics simulations under applied forces such as wind and gravity, enabling interactive and visually plausible 3D scene animations.

## 2 RELATED WORK

**2D World Models**     Some early works (Bell et al., 2015; Bear et al., 2021) learn to predict material labels on 2D images. Recently, learning forward dynamics from 2D video frames has also been explored extensively. For instance, Google's Genie (Parker-Holder et al., 2024) trains a next-frame prediction model conditioned on latent actions derived from user inputs, capturing intuitive 2D physics in an unsupervised manner. While these methods achieve impressive 2D generation and control, they do not explicitly model 3D geometry or a physically grounded world. Other works such as (Chen et al., 2024; Li et al., 2024) also explore generating or editing images based on learned real-world dynamics. While these methods achieve impressive results in 2D visual synthesis, they typically do not explicitly model 3D geometry, nor do they infer physically grounded material properties decoupled from appearances. These can lead to problems such as a lack of object permanence or implausible interactions. In contrast, PIXIE directly operates in 3D, predicting explicit physical parameters (e.g., Young's modulus, density) for 3D objects, enabling their integration into 3D physics simulators or neural networks (Wang et al., 2025; Mittal et al., 2025) for realistic interaction.

**Manual Assignment or Assignment of Physics using LLMs**     A number of recent methods have explored combining learned 3D scene representations (e.g., Gaussian splatting) with a physics solver where material parameters are assigned manually or through high-level heuristics. This often involves users specifying material types for the scene (Xie et al., 2023; Abou-Chakra et al., 2024) or using scripted object-to-material dictionaries (Qiu et al., 2024) or large language and vision-language models (Hsu et al., 2024; Chen et al., 2025a; Zhai et al., 2024; Le et al., 2024; Xia et al., 2024; Li et al., 2025; Cao et al., 2025; Shuai et al.) to guide the assignment.

**Test-time material optimization using videos**     Other works explore more automatic and principled ways to infer material properties using rendered videos. Some techniques (Jatavallabhula et al., 2021; Li et al., 2023; Zhong et al., 2024; Jiang et al., 2025; Zhang et al., 2025; Zhu et al., 2024) optimize material parameters by comparing simulated deformations against ground-truth observations, often requiring ground-truth multi-view videos of objects or particle positions under known forces. More recent approaches (Huang et al., 2024; Lin et al., 2025; Zhang et al., 2024) use video diffusion models as priors to optimize physics via a motion distillation loss. Notably, these approaches suffer from extremely slow per-scene optimization, often taking hours on a single scene, and do not generalize to new scenes. In contrast, PIXIE employs a feed-forward neural network that, once trained, predicts physical parameters in seconds, and can generalize to unseen scenes. A recent work Vid2Sim (Chen et al., 2025b) also aims to learn a generalizable material prediction network across scenes. This was done by encoding a front-view video of the object in motion with a foundation video transformer (Tong et al., 2022) and learning to regress these motion priors into physical parameters. Unlike Vid2Sim, PIXIE does not require videos, relying instead on visual features from static images. Overall, PIXIE can also be used as an informed warm-start along with these test-time methods to further refine predictions.

## 3 METHOD

Our central thesis is that 3D visual appearance provides sufficient information to recover an object's physical parameters. Texture, shading, and shape features captured from multiple calibrated images correlate with physical quantities such as Young's modulus and Poisson's ratio. By learning a mapping from these visual features to material properties, we can augment a volumetric reconstruction model (e.g., Gaussian Splatting) with a point-wise material estimate, without requiring force response observations. In Sec. 3.1, we detail our framework, leveraging rich visual priors from CLIP to predict a material field, which can be used by a physics solver to animate objects responding to external

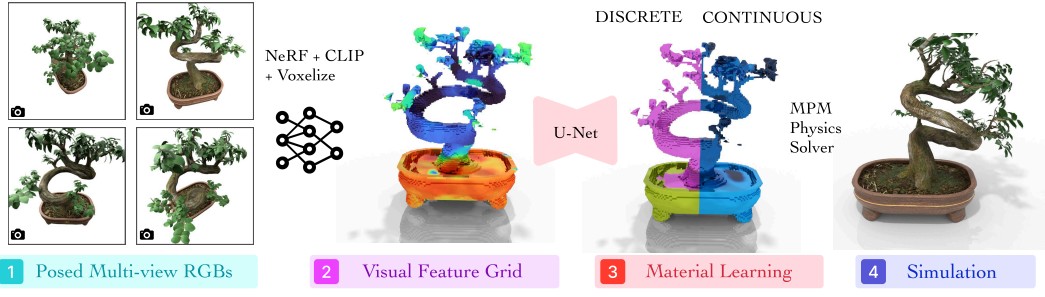

Figure 2: **Method Overview**. From posed multi-view RGB images of a static scene, PIXIE first reconstructs a 3D model with NeRF and distilled CLIP features (Shen et al., 2023). Then, we voxelize the features into a regular $N \times N \times N \times D$ grid where $N$ is the grid size and $D$ is the CLIP feature dimension. A U-Net neural network (Dhariwal and Nichol, 2021) is trained to map the feature grid to the material field $\hat{\mathcal{M}}_G$ which consists of a discrete material model ID and continuous Young's modulus, Poisson's ratio, and density value for each voxel. Coupled with a separately trained Gaussian Splatting model, $\hat{\mathcal{M}}_G$ can be used to simulate physics with a physics solver such as MPM.

forces. To train this model, we curated PIXIEVERSE, a large dataset of paired 3D assets and material annotations, as detailed in Sec. 3.2. Figure 2 gives an overview of our method.

## 3.1 PIXIE PHYSICS LEARNING

**Problem Formulation**   Formally, the goal is to learn a mapping:

$$f_\theta : (\mathcal{I}, \Pi) \longrightarrow \hat{\mathcal{M}} \tag{1}$$

that turns some calibrated RGB images of the static scene $\mathcal{I} = \{I_k\}_{k=1}^K$ and their joint camera specification $\Pi$ into a continuous three-dimensional *material field*. For every point $\mathbf{p} \in \mathbb{R}^3$ within the scene bounds, the field returns

$$\hat{\mathcal{M}}(\mathbf{p}) = \left( \hat{\ell}(\mathbf{p}), \ \hat{E}(\mathbf{p}), \ \hat{\nu}(\mathbf{p}), \ \hat{d}(\mathbf{p}) \right) \ ,$$

where $\hat{\ell} : \mathbb{R}^3 \to \{1, \dots, L\}$ is the discrete material class and $\hat{E}, \hat{\nu}, d : \mathbb{R}^3 \to \mathbb{R}$ are the continuous Young's modulus, Poisson's ratio, and density value respectively. Recall that the discrete material class, also known as the constitutive law, in Material Point Method is a combination of the choices of an expert-defined hyperelastic energy function $\mathcal{E}$ and return mapping $\mathcal{P}$ (Sec. A). Learning a point-mapping like this provides a fine-grained material segmentation where for every spatial location we assign both a semantic material label and the physical parameters that characterise that material. Learning the mapping in equation 1 directly from 2D images to 3D materials is not simple nor sample efficient. Instead, we leverage a distilled feature field which has rich visual priors to represent the intermediate mapping between 2D images and 3D visual features, and then a separate U-Net architecture to compute the mapping between 3D visual features and physical materials.

**3D Visual Feature Distillation**   Recent work on distilled feature fields has shown that dense 2D visual feature embeddings extracted from foundation models, such as CLIP, based on images can be lifted into 3D, yielding a volumetric representation that is both geometrically accurate and rich in terms of visual and semantic priors (Shen et al., 2023). Here, we also augment the classical NeRF representation (Mildenhall et al., 2021) to predict a view-independent feature vector in addition to color and density, i.e.,

$$F_\theta : (\mathbf{x}, \mathbf{d}) \mapsto \left( \mathbf{f}(\mathbf{x}), \ c(\mathbf{x}, \mathbf{d}), \ \sigma(\mathbf{x}) \right),$$

where $c \in \mathbb{R}^3$ and $\sigma \in \mathbb{R}_{\geq 0}$ are standard color and radiance NeRF outputs and $\mathbf{f} \in \mathbb{R}^d$ is a high-dimensional descriptor capturing visual semantics (e.g., object identity or other attributes), which we assume to be view-independent. We supervise color with image RGB and features with per-pixel CLIP embeddings extracted from the training images, using standard volume rendering (App. A.2). After training, we voxelize the feature field within known scene bounds to obtain a regular grid $F_G$ of dimension $N \times N \times N \times D$ grid, where $N = 64$ is the grid size and $D = 768$ is the CLIP feature dimension, serving as input to our material network.

**Material Grid Learning**   Our material learning network $f_M$ consists of a feature projector $f_P$ and a U-Net $f_U$. As the CLIP features are very high-dimensional, we learn a feature projector network $f_P$, which consists of three layers of 3D convolution mapping CLIP features $\mathbb{R}^{768}$ to a

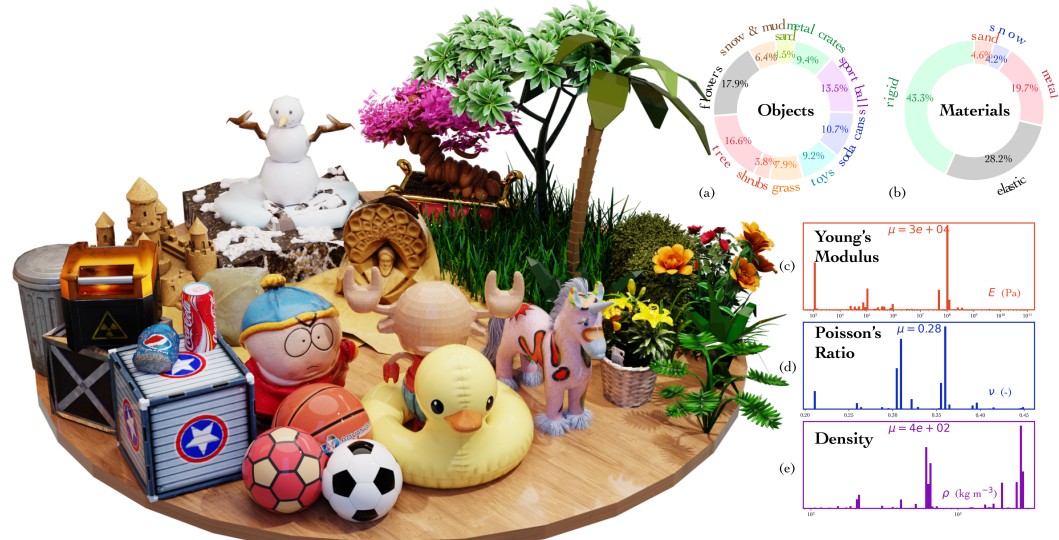

Figure 3: **PIXIEVERSE Dataset Overview.** We collect 1624 high-quality single-object assets, spanning 10 semantic classes (a), and 5 constitutive material types (b). The dataset is annotated with detailed physical properties including discrete material types (b), Young's modulus (c), Poisson's ratio (d), and mass density (e). The left figure shows representative examples from the dataset: organic matter (*tree, shrubs, grass, flowers*), deformable toys (*rubber ducks*), sports equipment (*sport balls*), granular media (*sand, snow & mud*), and hollow containers (*soda cans, metal crates*).

low-dimensional manifold $\mathbb{R}^{64}$. We then use the U-Net architecture $f_U$ to learn the mapping from the projected feature grid $F_G$ to a material grid $\hat{\mathcal{M}}_G(\mathbf{p})$, which is a voxelized version of the material field $\hat{\mathcal{M}}(\mathbf{p})$. The feature projector $f_P$ and U-Net $f_U$ are jointly trained end-to-end via a cross-entropy and mean-squared error loss to predict the discrete material classification and the continuous values including Young's modulus, Poisson's ratio and density. More details is in Appendix E.

We found that our voxel grids are very sparse with around 98% of the voxels being background. Naively trained, the material network $f_M$ would learn to always predict background. Thus, we also separately compute an occupancy mask grid $\mathbb{M} \in \mathbb{R}^N \times \mathbb{R}^N \times \mathbb{R}^N$, constructed by filtering out all voxels whose NeRF densities fall below a threshold $\alpha = 0.01$. The supervised losses—cross-entropy and mean squared errors—are only enforced on the occupied voxels. Concretely, the masked supervised loss consists of a discrete cross-entropy and continuous mean-squared error loss:

$$\mathcal{L}_{\text{sup}} = \frac{1}{N_{occ}} \sum_{\mathbf{p} \in \mathcal{G}} \mathbb{M}(\mathbf{p}) \Big[ \lambda \cdot \text{CE}(\hat{\ell}(\mathbf{p}), \ell^{GT}(\mathbf{p})) + (\hat{E}(\mathbf{p}) - E^{GT}(\mathbf{p}))^2$$
$$+ (\hat{\nu}(\mathbf{p}) - \nu^{GT}(\mathbf{p}))^2 + (\hat{d}(\mathbf{p}) - d^{GT}(\mathbf{p}))^2 \Big] \ , \tag{2}$$

where $N_{occ} = \sum_{\mathbf{p} \in \mathcal{G}} \mathbb{M}(\mathbf{p})$ is the total number of occupied voxels in the grid, $\hat{\ell}(\mathbf{p})$ and $\ell^{GT}(\mathbf{p})$ are the predicted material class logits and the ground-truth, $CE$ is the cross-entropy loss, $\lambda$ is a loss balancing factor, and $E, \nu, d$ are the Young's modulus, Poisson's ratio and density values, respectively.

**Physics Simulation** We use the Material Point Method (MPM) to simulate physics. The MPM solver (Sec. A.3) takes a point cloud of initial particle poses along with predicted material properties, and the external force specification, and simulates the particles' transformations and deformations. Following PhysGaussian (Xie et al., 2023), we learn a 3DGS model incorporating techniques such as internal fillings, making it suitable for MPM simulation. We then transfer the material properties from our predicted material grid into the 3DGS model via nearest neighbor interpolation.

### 3.2 PIXIEVERSE DATASET

We collect one of the largest and highest quality known datasets of diverse objects with annotated physical materials. Our dataset (Fig. 3) covers 10 semantic classes, ranging from organic matter (trees, shrubs, grass, flowers) and granular media (sand, snow and mud) to hollow containers (soda-cans, metal crates), and toys (rubber ducks, sport balls). The dataset is sourced from Objaverse (Deitke et al., 2022). Since Objaverse objects do not have physical parameter annotations, we develop an

semi-automatic multi-stage labeling pipeline leveraging foundation vision-language models i.e., Gemini-2.5-Pro (Team et al., 2023), distilled CLIP feature field (Kobayashi et al., 2022) and manually tuned in-context physics examples. The full details is given in Appendix B and C.

## 4 EXPERIMENTS

**Dataset**    We train PIXIE on the PIXIEVERSE dataset and evaluate on 38 synthetic scenes from the test set and six real-world scene from the NeRF (Mildenhall et al., 2021), LERF (Kerr et al., 2023) and Spring-Gaus (Zhong et al., 2024) datasets.

**Simulation Details**    We use the material point method (MPM) implementation from PhysGaussian (Xie et al., 2023) as the physics solver. The solver takes a gaussian splatting model augmented with physics where each Gaussian particle also has a discrete material model ID, and continuous Young's modulus, Poisson's ratio, and density values. Each simulation is run for around 50 to 125 frames on a single NVIDIA RTX A6000 GPU. External forces such as gravity and wind are applied to the static scenes as boundary conditions to create physics animations.

**Baselines**    We evaluate PIXIE against two recent test-time optimization methods: DreamPhysics (Huang et al., 2024) and OmniPhysGS (Lin et al., 2025), and a LLM method – NeRF2Physics (Zhai et al., 2024). DreamPhysics optimizes a Young's modulus field, requiring users to specify other values including material ID, Poisson's ratio, and density. OmniPhysGS, on the other hand, selects a hyperelastic energy density function and a return mapping model, which, in combination, specifies a material ID for each point in the field, requiring other physics parameters to be manually specified. Both methods rely on a user prompt such as "a tree swing in the wind" and a generative video diffusion model to optimize a motion distillation loss. PIXIE, in contrast, infers all discrete and continuous parameters jointly (Fig. 16). NeRF2Physics first captions the scene and queries a LLM for all plausible material types (e.g., "metal") along with the associated continuous values. Then, the material semantic names are associated with 3D points in the CLIP feature field, and physical properties are thus assigned via weighted similarities. This method is similar to our dataset labeling in principle with some crucial differences as detailed in Appendix B and C, allowing PIXIEVERSE to have much more high-quality labels. PIXIE was trained on 12 NVIDIA RTX A6000 GPUs, each with a batch size of 4, in one day using the Adam optimizer (Kingma, 2014) while prior test-time methods do not require training. For training PIXIE and computing metrics, we apply a log transform to $E$ and $\rho$, and normalize all $\log E, \nu, \log \rho$ values to $[-1, 1]$ based on max/min statistics from PIXIEVERSE.

**Evaluation Metrics**    We utilize a state-of-the-art vision-language model, Gemini-2.5-Pro (Team et al., 2023) as the judge. The models are prompted to compare the rendered candidate animations generated using physics parameters predicted by different baselines, and score those videos on a scale from 0 to 5, where a higher score is better. The prompt is in Appendix D. We also measure the reconstruction quality using PSNR and SSIM metric against the reference videos in the PIXIEVERSE dataset, which are manually verified by humans for quality control. Other metrics our method optimizes including class accuracy and continuous errors over $E, \nu, \rho$ are also computed.

### 4.1 SYNTHETIC SCENE EXPERIMENTS

Figure 4 (a) plots Gemini score versus runtime. PIXIE achieves a VLM realism score of **4.35 ± 0.08** – a **1.46-4.39x** improvement over all baselines and tops all other metrics – while reducing inference time from minutes or hours to **2 s**. A A per-class breakdown in Fig. 4 (b) shows our lead in most classes. In Table 1, our model improves perceptual metrics such as PSNR and SSIM by $3.6 - 30.3\%$ and VLM scores by $2.21 - 4.58x$ over prior works. Figure 5 visualises eight representative scenes, comparing PIXIE against prior works. DreamPhysics leaves stiff artifacts due to missegmentation or overly high predicted $E$ values, OmniPhysGS collapses under force, and NeRF2Physics introduces high-frequency noise, whereas PIXIE generates smooth, class-consistent motion and segment boundaries. In the appendix, Figure 16 qualitatively visualizes the physical properties predicted by our network, showing PIXIE's ability to cleanly and accurately recover both discrete and continuous parameters across a diverse sets of objects and continuous value spectrum. In contrast, some prior methods can only recover a subset of parameters like $E$ or material class. The end-to-end runtime is also included in Tab. 6. Additionally, we conducted a blind user study comprising of 512 responses from human volunteers who was asked to rank the realism of different methods. The result is included in the Appendix Tab. 4. The volunteers rank PIXIE higher than the next best method about 20% more often.

Table 1: **Main Quantitative Results.** We report the average reconstruction quality (PSNR, SSIM) against the reference videos in PIXIEVERSE, the VLM score, and five other metrics our method optimizes including material accuracy and continuous errors over $E, \nu, \rho$. Standard errors and 95% CI are also included, and best values are **bolded**. PIXIE-CLIP is by far the best method across all metrics, achieving 1.62-5.91x improvement in VLM score and 3.6-30.3% gains in PSNR and SSIM. Our CLIP variant is also notably more accurate than RGB and occupancy features as measured by material class accuracy and average continuous MSE on the test set. While our method simultaneously recovers all physical properties, some prior works only predict a subset, hence - (empty) entries.

| Method | PSNR ↑ | SSIM ↑ | VLM ↑ | Mat. Acc.↑ | Avg. Cont. MSE ↓ | log $E$ err ↓ | $\nu$ err ↓ | log $\rho$ err ↓ |
|---|---|---|---|---|---|---|---|---|
| DreamPhysics (Huang et al., 2024) | | | | | | | | |
| 1 epoch | 19.398±1.090 | 0.880±0.020 | 2.97±0.31 | - | - | 2.393±0.123 | - | - |
| 25 epochs | 19.078±0.939 | 0.881±0.019 | 2.68±0.24 | - | - | 1.419±0.097 | - | - |
| 50 epochs | 19.189±0.980 | 0.880±0.020 | 2.53±0.24 | - | - | 1.387±0.097 | - | - |
| OmniPhysGS (Lin et al., 2025) | | | | | | | | |
| 1 epoch | 17.907±0.359 | 0.882±0.007 | 0.74±0.10 | 0.072±0.0511 | - | - | - | - |
| 2 epochs | 17.889±0.372 | 0.882±0.007 | 1.23±0.19 | 0.109±0.0704 | - | - | - | - |
| 5 epochs | 17.842±0.354 | 0.883±0.007 | 0.99±0.12 | 0.104±0.0681 | - | - | - | - |
| NeRF2Physics (Zhai et al., 2024) | 18.517±0.644 | 0.886±0.013 | 1.09±0.28 | 0.274±0.001 | 0.858±0.109 | 1.115±0.165 | 0.462±0.106 | 0.997±0.162 |
| PIXIE | | | | | | | | |
| Occupancy | 17.887±1.524 | 0.866±0.027 | 1.76±0.41 | 0.643±0.052 | 0.126±0.012 | 0.149±0.023 | 0.124±0.014 | 0.105±0.015 |
| RGB | 18.652±2.031 | 0.861±0.035 | 2.53±0.46 | 0.722±0.061 | 0.106±0.015 | 0.196±0.032 | 0.079±0.012 | 0.045±0.014 |
| **CLIP (ours)** | **23.256±2.456** | **0.918±0.023** | **4.35±0.08** | **0.985±0.011** | **0.056±0.005** | **0.022±0.004** | **0.034±0.006** | **0.112±0.009** |

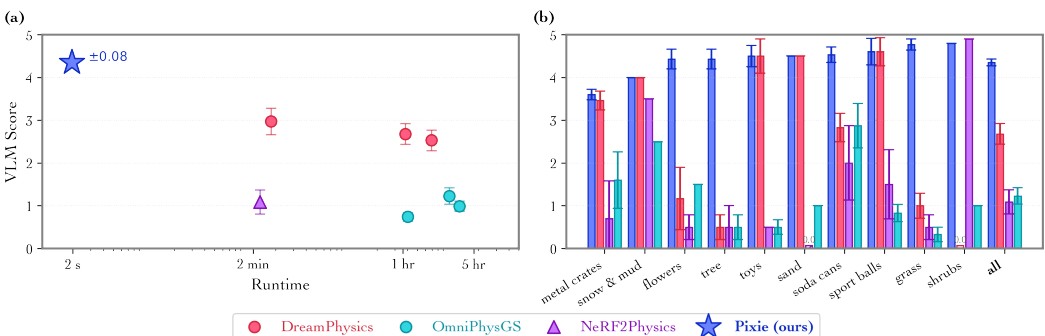

Figure 4: **Main VLM Results.** (a) **VLM score versus wall-clock time:** PIXIE is three orders of magnitude faster than previous works while achieving 1.46-4.39x improvement in realism. Test-time optimization methods are run with varying numbers of epochs i.e., $1, 25, 50$ for DreamPhysics and $1, 2, 5$ for OmniPhysGS while inference methods are only run once. (b) **Per-class VLM score:** Our method leads on most object classes. Standard errors are also included.

## 4.2 ZERO-SHOT GENERALIZATION TO REAL-WORLD SCENES

Without any real-scene supervision, PIXIE can zero-shot generalize to many real-world scenes as shown in Fig. 6. For example, our method correctly assigns rigid vase bases and flexible leaves, yielding realistic motion that closely matches human expectation. Our method is surprisingly performant despite significant and non-trivial visual gaps between the training synthetic data versus the out-of-distribution real-world scenes. No other baseline can generalize under this setting. The quantitative reconstruction performance is included in the Appendix Tab. 3.

To test the generalizability of PIXIE to more diverse and out-of-distribution real-world objects, we also conduct a quantitative experiment on the ABO-500 dataset (Zhai et al., 2024). This dataset contains 500 real-world objects sourced from products sold on Amazon. The mass estimation performance is reported in Appendix Sec. H. Despite never trained on categories such as household furniture, our method can produce reasonable predictions, outperforming other baselines.

## 4.3 PIXIE'S FEATURE TYPE ABLATION

Replacing CLIP with RGB or occupancy features drops VLM score by 40-60 % and nearly doubles parameter MSE (Table 1, rows "Occupancy" and "RGB"). We provide more results in the Appendix. Specifically, we show that the material class prediction also dramatically drops across all classes as shown in Fig. 17. Figure 18 shows the failure modes for real scenes, highlighting RGB and occupancy's struggle to generalize to unseen data as compared to CLIP.

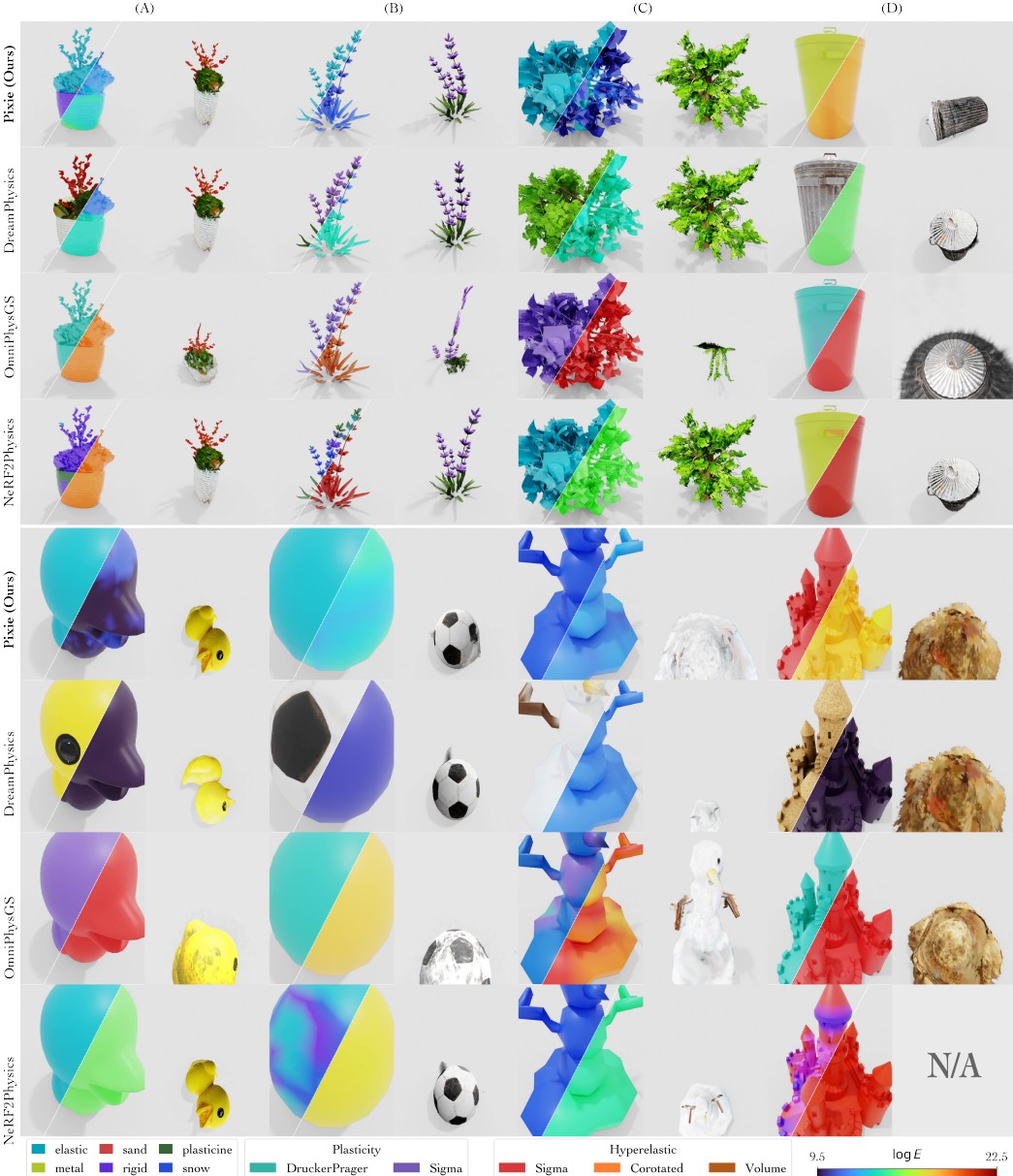

Figure 5: **Qualitative comparison on synthetic scenes.** We visualized the predicted material class and $E$ predictions (left, right respectively) for PIXIE and Nerf2Physics, $E$ for DreamPhysics (right), and the plasticity and hyperelastic function classes predicted by OmniPhysGS. PIXIE produces stable, physically plausible motion while DreamPhysics remains overly stiff due to inaccurate fine-grained $E$ prediction or too high $E$ (e.g., see tree (C)), OmniPhysGS collapses under load due to unrealistic combination of plasticity and hyperelastic functions, and NeRF2Physics exhibits noisy artifacts. Please see https://pixie-2026-12998.github.io/ for the videos.

## 5    CONCLUSION AND LIMITATIONS

We presented PIXIE, a framework that jointly reconstructs geometry, appearance, and explicit physical material fields from posed RGB images. By distilling rich CLIP features into 3D and training a feed-forward 3D U-Net with per-voxel material supervision on our new PIXIEVERSE dataset, PIXIE avoids the expensive test-time optimization required by prior work. Once trained, it produces full material fields in a few seconds, improving Gemini realism scores by 1.46-4.39x over prior art while reducing inference time by three orders of magnitude. PIXIE leverages CLIP's strong visual priors, which enables zero-shot transfer to real scenes, even though it is only trained on synthetic data. The method enables realistic, physically plausible 3D scene animation with off-the-shelf MPM solvers.

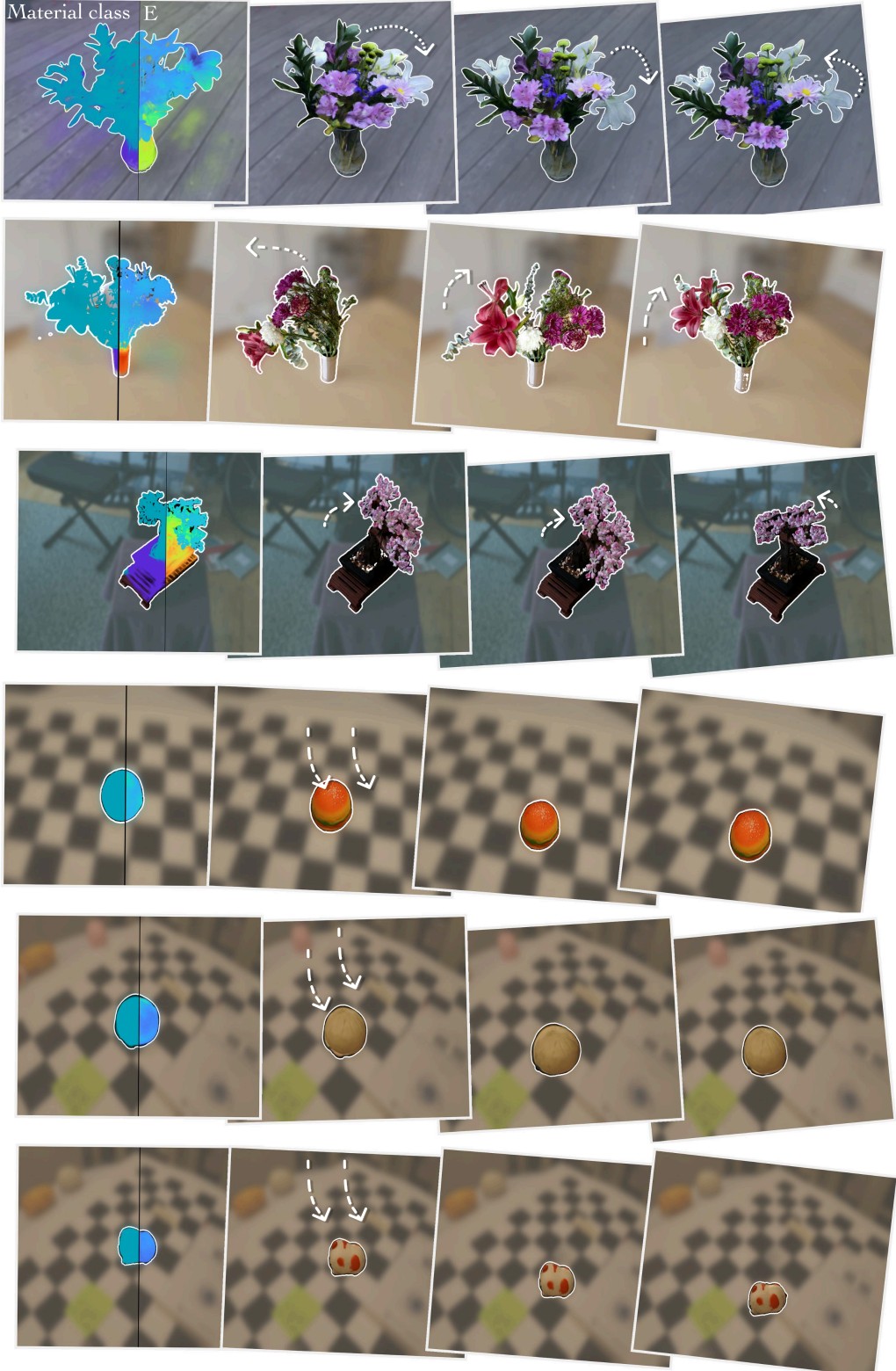

Figure 6: **PIXIE's Zero-shot Real-scene Generalization.** Trained only on synthetic PIXIEVERSE, PIXIE can predict plausible physic properties, enabling realistic MPM simulation of real scenes. Here, we visualize the material types (left) and Young's modulus (right) prediction in the first frame, and subsequent frames impacted by a wind force. Please see the videos in our website https://pixie-2026-12998.github.io/.

**Limitations and Future Works**    We take the first step towards learning a supervised 3D model for physical material prediction. Like prior art, our work focuses on single object interactions leaving multi-object scenes for future investigation. Another limitation is that while our feed-forward model predicts a point estimate for each voxel, materials in the real-world contain uncertainty that visual information alone cannot resolve (e.g., a tree can be stiff or flexible). A promising extension is to learn a distribution of materials (e.g., using diffusion) instead. Nonetheless, the current PIXIE model which performs inference on static scenes can serve as a powerful prior for further fine-tuning via test-time optimization methods as shown in Appendix Sec. G,

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

# A PRELIMINARIES

This section briefly reviews foundational concepts in 3D scene representation and physics modeling relevant to our work.

## A.1 LEARNED SCENE REPRESENTATION

Reconstructing 3D scenes from 2D images is commonly achieved by learning a parameterized representation, $F_\theta$, optimized to render novel views that match observed images $\{I^{(i)}\}_{i=1}^M$ given camera parameters $\{\pi^{(i)}\}_{i=1}^M$. This typically involves minimizing a photometric loss:

$$\min_\theta \sum_{i=1}^M \left\| \hat{I}^{(i)}(\theta) - I^{(i)} \right\|_2^2 ,$$

where $\hat{I}^{(i)}(\theta)$ is the image rendered from viewpoint $i$. Two prominent representations are Neural Radiance Fields (NeRF) and Gaussian Splatting (GS) models.

**Neural Radiance Fields (NeRF)** (Mildenhall et al., 2021) model a scene as a continuous function $F_\theta : (\mathbf{x}, \mathbf{d}) \mapsto (c, \sigma)$, mapping a 3D location $\mathbf{x}$ and viewing direction $\mathbf{d}$ to an emitted color $c$ and volume density $\sigma$. Images are synthesized using volume rendering, integrating color and density along camera rays. This process' differentiability allows for end-to-end optimization from images.

**Gaussian Splatting (GS)** (Kerbl et al., 2023) represents scenes as a collection of 3D Gaussian primitives, each defined by a center $\mu_i$, covariance $\Sigma_i$, color $\mathbf{c}_i$, and opacity $\alpha_i$. These Gaussians are projected onto the image plane and blended using alpha compositing to render views.

In our work, the principles of neural scene representation, particularly NeRF-like architectures, are leveraged not only for visual reconstruction but also for creating dense 3D visual feature fields. As detailed in Sec. 3.1, we utilize a NeRF-based model to distill 2D image features (e.g., from CLIP) into a volumetric 3D feature grid. This 3D feature representation, $F_G$, then serves as the primary input to our physics prediction network. For subsequent physics simulation, GS offers a convenient particle-based representation.

## A.2 3D VISUAL FEATURE DISTILLATION DETAILS

Following (Shen et al., 2023), we augment the NeRF mapping to produce features $\mathbf{f}$ alongside color $c$ and density $\sigma$:

$$F_\theta : (\mathbf{x}, \mathbf{d}) \mapsto \big( \mathbf{f}(\mathbf{x}),\ c(\mathbf{x}, \mathbf{d}),\ \sigma(\mathbf{x}) \big).$$

Given a camera ray $r(t) = \mathbf{o} + t\mathbf{d}$ passing through pixel $p$, color $C(p)$ and features $F(p)$ are volume-rendered as

$$C(p) = \int_{t_n}^{t_f} T(t)\, \sigma\big(r(t)\big)\, c\big(r(t), \mathbf{d}\big)\, dt, \qquad F(p) = \int_{t_n}^{t_f} T(t)\, \sigma\big(r(t)\big)\, f\big(r(t)\big)\, dt, \qquad (3)$$

where $T(t) = \exp\big(-\int_{t_n}^t \sigma(r(s))\, ds\big)$ is the accumulated transmittance from the ray origin to depth $t$. At each training iteration, a batch of rays is sampled from the input views. For each ray $r$ (pixel $p$), we enforce that the rendered color $C(p)$ matches the ground-truth pixel RGB $C^*(p)$, while the rendered feature $F(p)$ matches the corresponding CLIP-based feature vector $F^*(p)$ extracted from the image. The loss of the network is:

$$\mathcal{L} = \sum_p \big\| C(p) - C^*(p) \big\|_2^2 + \lambda_{\text{feat}} \sum_p \big\| F(p) - F^*(p) \big\|_2^2 ;$$

the first term enforces color fidelity, while the second aligns the rendered volumetric CLIP features with the dense 2D features extracted from the training images.

From a trained distilled feature field $F_\theta$, we obtain a regular feature grid $F_G$ of dimension $N \times N \times N \times D$ grid, where $N = 64$ is the grid size and $D = 768$ is the CLIP feature dimension. This is done via voxelization using known scene bounds. For our synthetic dataset, we center and normalize all objects within a unit cube.

## A.3 MATERIAL POINT METHOD (MPM) FOR PHYSICS SIMULATION

To simulate how objects move and deform under applied forces, a physics engine requires knowledge of their material properties. These properties are typically defined within the framework of continuum mechanics, which describes the behavior of materials at a macroscopic level. The fundamental equations of motion (conservation of mass and momentum) are:

$$\rho\frac{D\mathbf{v}}{Dt} = \nabla \cdot \boldsymbol{\sigma} + \mathbf{f}^{\text{ext}} \qquad\qquad \nabla \cdot \mathbf{v} = 0 \ , \tag{4}$$

where $\rho$ is mass density, $\mathbf{v}$ the velocity field, $\boldsymbol{\sigma}$ the Cauchy stress tensor, and $\mathbf{f}^{\text{ext}}$ any external force (e.g. gravity or user interactions). The material-specific *constitutive laws* define how $\boldsymbol{\sigma}$ depends on the local deformation gradient $\mathbf{F}$. For elastic materials, stress depends purely on the recoverable strain; for plastic materials, a yield condition enforces partial "flow" once strain exceeds a threshold.

**Constitutive Laws and Parameters**   Most continuum simulations separate the constitutive model into two core components:

$$\begin{aligned} \mathcal{E}_\mu &: \mathbf{F}^e \ \mapsto \ \mathbf{P}, \\ \mathcal{P}_\mu &: \mathbf{F}^{e,\text{trial}} \ \mapsto \ \mathbf{F}^{e,\text{new}} \ , \end{aligned} \tag{5}$$

where $\mathbf{F}^e$ is the *elastic* portion of the deformation gradient, $\mathbf{P}$ is the (First) Piola–Kirchhoff stress, and $\mu$ represents the set of material parameters (e.g. Young's modulus $E$, Poisson's ratio $\nu$, yield stress). The *elastic law* $\mathcal{E}_\mu$ computes stress from the current elastic deformation, while the *return-mapping* $\mathcal{P}_\mu$ projects any "trial" elastic update $\mathbf{F}^{e,\text{trial}}$ onto the feasible yield surface if plastic flow is triggered. Typically, the constitutive laws i.e., $\mathcal{E}_\mu$ and $\mathcal{P}_\mu$ are hand-designed by domain experts. The choice of $\mathcal{E}$ and $\mathcal{P}$ jointly define a class of material (e.g., rubber). Within a material class, additional continuous parameters $\mu$ including Young's modulus, Poisson's ratio and density can be specified for a more granular control of the material properties (e.g., stiffness of rubber). In our work, PIXIE jointly predicts the discrete material model and the continuous material parameters.

## B   PIXIEVERSE DATASET DETAILS

We heavily curate the dataset to a set of $1624$ objects after a multi-stage filter that removes multi-object scenes, missing textures, duplicated assets, and objects whose material labeling is either ambiguous or physically implausible. The process is semi-automatic with a VLM-driven multi-stage pipeline while still imparting substantial human prior and labor. We manually tune the physics parameter ranges for each semantic class (e.g., "tree", "rubber toy") and 3D segmentation query terms, and provide these as in-context examples for the VLM to align them with human's physical understanding.

First, we define some object class (e.g., "tree") and some alternative query terms (e.g., "ficus, fern, evergreen etc"). We then use a sentence transformer model (Wang et al., 2020) to compute the cosine similarity between the search terms and the name of each Objaverse object. We select $k = 500$ objects with the highest similarity score for each class, creating an initial candidate pool. However, since Objaverse objects vary greatly in asset quality, lighting conditions, and some scenes contain multiple objects which are not suitable for our material learning, an additional filtering step is needed. The Gemini VLM is prompted to filter out low-quality or unsuitable scenes. A distilled NeRF model is fitted to each object. Then, the VLM is provided five multi-view RGB images of an object, and prompted to provide a list of the object's semantic parts along with associated material class and ranges for continuous values. The ranges such as $E \in \{1e4, 1e5\}$ allow us to simulate a wider range of dynamics from flexible to more rigid trees. The VLM is also prompted to specify a list of constraints such as to ensure that the leaf's density is lower than the trunk's. We then sample the continuous values from the VLM's specified ranges subject to the constraint via rejection sampling. The semantic parts (e.g., "pot") are used with the CLIP distilled feature field to compute a 3D semantic segmentation of the object into parts, and the sampled material properties are applied uniformly to all points within a part. This ground-truth material and feature fields are then voxelized into regular grids for use in supervised learning by the PIXIE framework.

The following sections provide more details on each step of our semi-automatic labeling process.

```
tree: tree, ficus, fern, oak tree, pine tree, evergreen, palm tree, maple tree,
bonsai tree
flowers: flower, bouquet, rose, tulip, daisy, lily, sunflower, orchid, flower
arrangement, flowering plant, garden flowers, wildflowers, floral
rubber_ducks_and_toys: rubber duck, bath toy, rubber toy, toy duck, squeaky toy,
floating toy, plastic duck, children's bath toy, yellow duck toy, rubber animal
toy
soda_cans: soda can, aluminum can, beverage can, cola can, soft drink can, metal
can, canned drink, pop can, fizzy drink can
sport_balls: basketball, soccer ball, football, tennis ball, baseball, volleyball,
 golf ball, rugby ball, ping pong ball, cricket ball, bowling ball, beach ball,
sports ball
sand: sand, beach sand, desert sand, sandy terrain, sand pile, sand dune, sandpit,
 sand box, sand texture, grainy sand
shrubs: shrub, bush, hedge, ornamental bush, garden shrub, boxwood, flowering
bush, evergreen shrub, decorative plant, landscaping shrub
metal_crates: metal crate, steel box, metal container, shipping crate, metal
storage box, industrial container, metal chest, storage crate, metallic box
grass: grass, lawn, turf, grassland, meadow, grassy field, green grass, grass
patch, tall grass, wild grass, pasture
snow_and_mud: snow, mud, snowy ground, muddy ground, wet mud, fresh snow, packed
snow, snowy terrain, muddy terrain, snow patch, mud puddle, snowdrift, muddy path,
 snowy surface, muddy surface, slush, wet snow, dirty snow, muddy water, snowy
landscape
```

Figure 7: **Objaverse Class Selection Keywords.** The keywords for matching a semantic class with an objaverse asset's name.

### B.1 OBJECT SELECTION FROM OBJAVERSE

We use the all-MiniLM-L6-v2 (Wang et al., 2020) sentence transformer to compute the cosine similarity between an objaverse asset's name and some search terms for each object class. The search terms are in Fig. 7. The top $k = 500$ objects with the highest similarity score are selected for each class.

### B.2 OBJECT FILTERING

Next, we prompt Gemini to filter out low-quality assets. The system instruction is given in Fig. 8. Then, a human quickly scans through the VLM results organized in our web interface as shown in Fig. 9 to correct any mistakes.

### B.3 CLIP-DRIVEN 3D SEMANTIC SEGMENTATION

From a distilled CLIP feature field of the object (Shen et al., 2023), we can perform 3D semantic segmentation by providing a list of the object's parts (e.g., "pot, trunk, leaves"). These query terms are used to compute the cosine similarity between each CLIP feature at a given 3D coordinate against the terms, and the part with highest similarity is assigned to that point. The choices of query terms (e.g., "pot, trunk, leaves" vs "base, stem, leaf") greatly affect the segmentation quality, and is not obvious. A high-performing query list in one object is not guaranteed to yield high performance in another object, e.g., see Fig. 10. Thus, we prompt a VLM actor to generate several candidate queries for each object, render all candidates, and prompt another VLM critic to select the best query terms from the rendered 3D segmentation images, as detailed Sec. B.4.

### B.4 VLM ACTOR-CRITIC LABELING

Current VLMs might not have robust physical understanding for generating high-quality labels for PIXIEVERSE zeroshot. Thus, we first manually tune the physic parameters for each semantic object

```
We need to select some images of the classes: {class_name}. This class includes
objects like {search_terms}. We will provide you some images rendered from the 3D
 model. You need to either return True or False. Return False to reject the image
 as inappropriate for the video game development. Some common reasons for
rejection:
  — The image doesn't clearly depict the object class
  — The image is too dark or too bright or too blurry or has some other low
quality.
    Remember, we want high—quality training data.
  — The image contains other things in addition to the object.
REMEMBER, we only want images that depict cleanly ONE SINGLE OBJECT belonging to
one of the classes. But you also need to use your common sense and best judgement.
 For example, for a class like "flowers", the object might include a vase of
flowers (you rarely see a single flower in the wild). So you should return True
in this case.
  — We do want diversity in our dataset collection. So even if the texture of the
 object is a bit unusual, as long as you can recognize it as belonging to the
class / search terms, you should return True. Only remove low—quality assets.

The return format is:
```json
{
  "is_appropriate": true (or false),
  "reason": "reason for the decision"
}
```
We'll be using the 3d models to learn physic parameters like material and young
modulus to simulate the physics of the object. E.g., the tree swaying in the wind
 or thing being dropped from a height. Therefore, you need to decide if the image
 depicts an object that is likely to be used in a physics simulation.
```

Figure 8: **Object Filtering System Prompt.** Prompt for VLM to filter out low-quality assets.

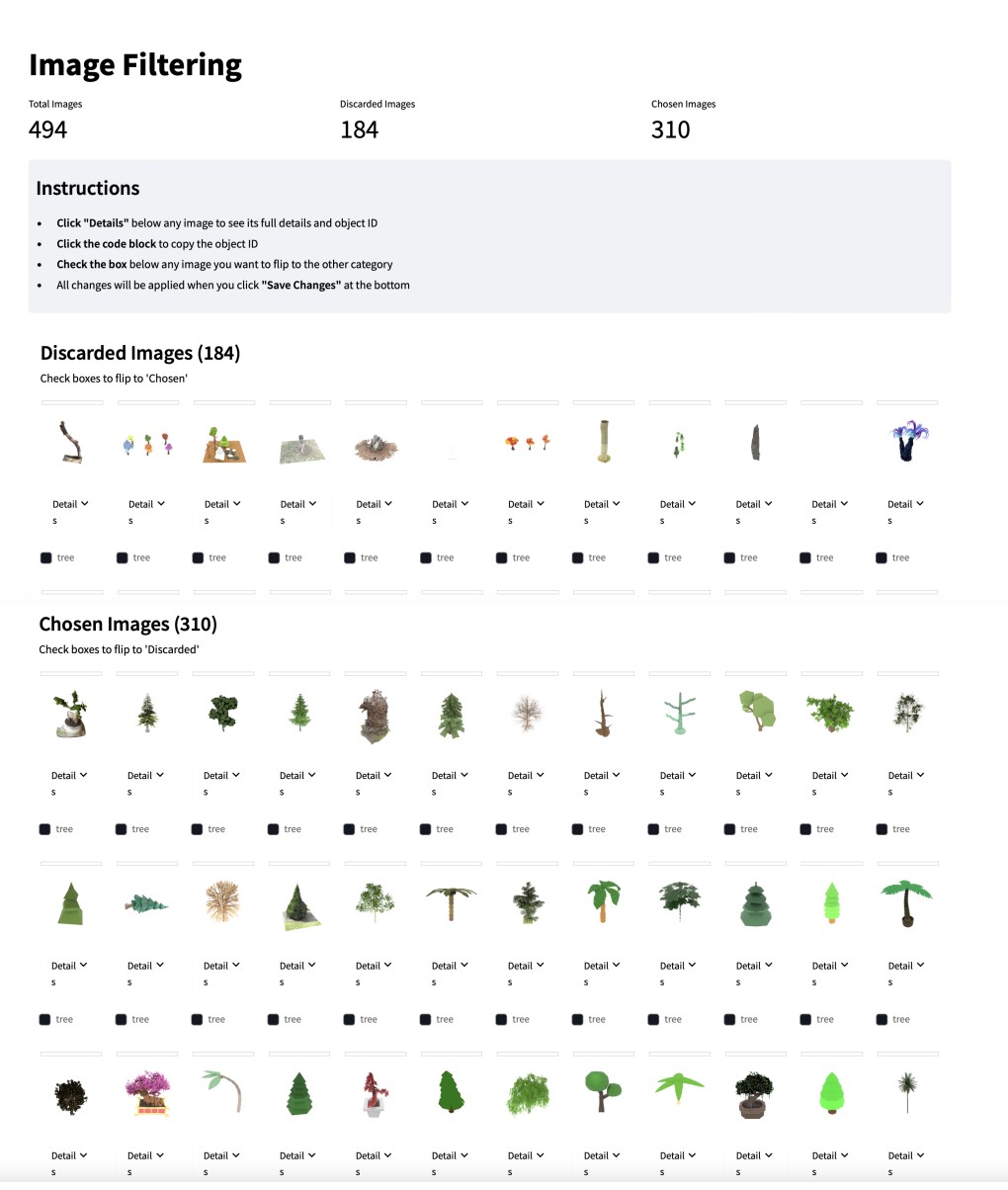

Figure 9: **Manual correction for object filtering.** The web interface for quickly inspecting and manually correcting VLM results.

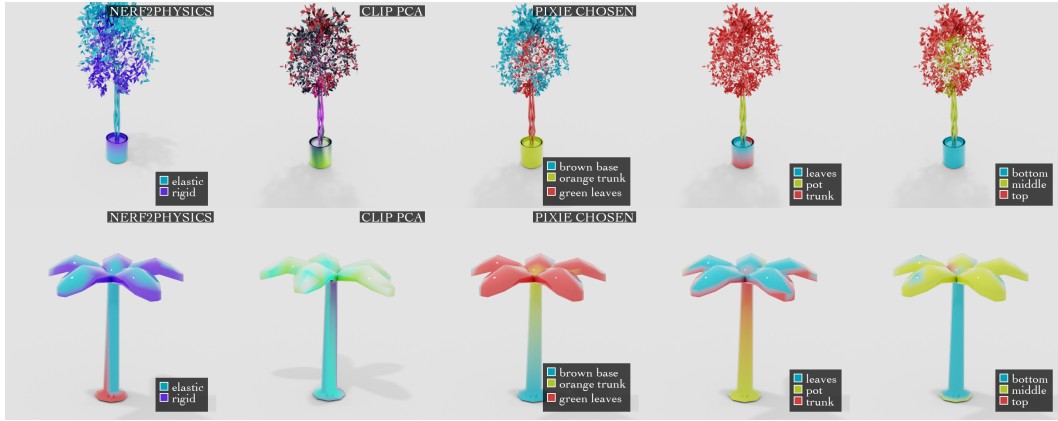

Figure 10: **CLIP Semantic Segmentation.** CLIP features can be noisy for various objects and different text queries vary greatly in segmentation quality. Thus, we prompt a VLM actor to generate several candidate queries for each object, render all candidates, and prompt another VLM critic to select the best query terms from the rendered 3D segmentation images. Some candidates are provided and proposals chosen by the critic are highlighted. Note that a high-performing query proposal (e.g., "leaves,pot,trunk") in one object is not necessary high-performant in another. The PCA visualization of the CLIP feature fields is also provided.

class (e.g., "tree", "rubber toy"). A condensed version of these examples is provided in Fig. 12. We also provide examples of different search terms (e.g., "pot, trunk, leaves" vs "base, stem, leaf"). These in-context examples are provided to a VLM actor that simultaneously proposes physics parameters and semantic segmentatic queries for that object from multi-view images of that object as illustrated in Fig. 11. The full system prompt for the VLM is provided in Fig. 13 and the full in-context examples in Listing 1. We render an image representing 3D semantic segmentation masks for each query proposal as shown in Fig. 10. A VLM critic is then prompted to select the best segmentation queries from the rendered images. The critic's system prompt is provided in Fig. 14.

Additionally, materials in the real-world contain uncertainty that visual information alone cannot resolve (e.g., a tree can range from stiff to flexible). Thus, instead of specifying one physics parameter per part, we prompt the VLM actor to output a plausible range (e.g., $E \in \{1e4, 1e5\}$ see Fig. 11, 12). We then sample a value uniformly from each range to build our training dataset. To further ensure that the sampled values are consistent, the VLM is also prompted to specify a list of constraints (e.g., the density of leaves must be lower than that of the trunk). Rejection sampling is used to ensure that the final dataset respects the constraints.

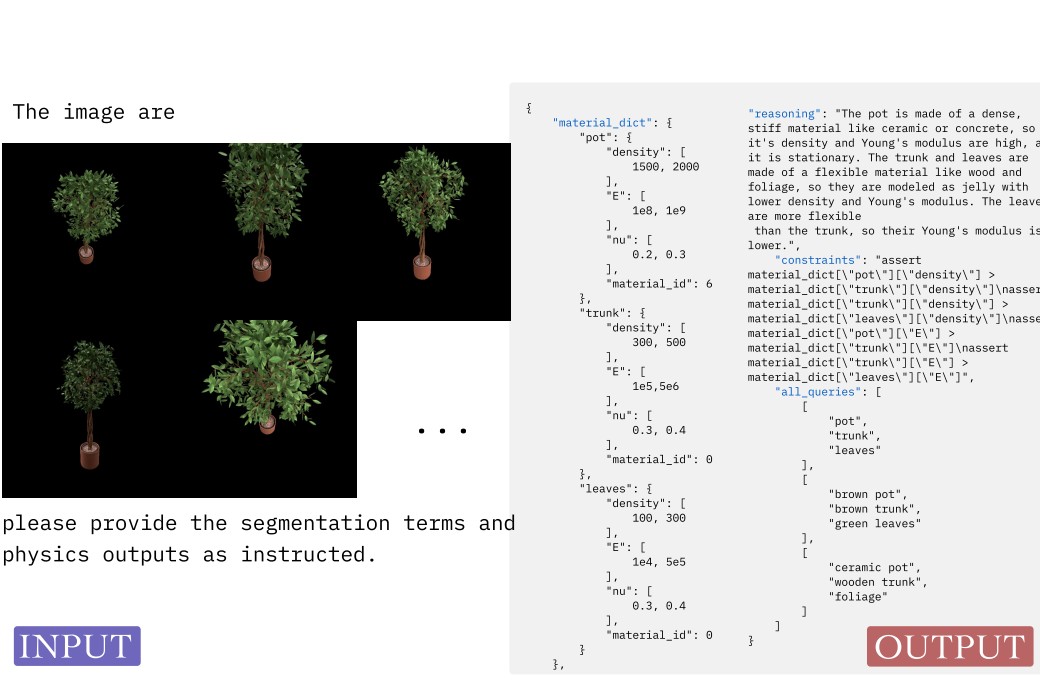

Figure 11: **VLM Actor's Physics and Segmentation Proposal.**

```
tree:
  pot: {density: 400, E: 2e8, nu: 0.4, material: "rigid"}
  trunk: {density: 400, E: 2e6, nu: 0.4, material: "elastic"}
  leaves: {density: 200, E: 2e4, nu: 0.4, material: "elastic"}

flowers:
  vase: {density: 500, E: 1e6, nu: 0.3, material: "rigid"}
  flowers: {density: 100, E: 1e4, nu: 0.4, material: "elastic"}

shrub:
  stems: {density: 300, E: 1e5, nu: 0.35, material: "elastic"}
  twigs: {density: 250, E: 6e4, nu: 0.38, material: "elastic"}
  foliage: {density: 150, E: 2e4, nu: 0.40, material: "elastic"}

grass:
  blades: {density: 80, E: 1e4, nu: 0.45, material: "elastic"}
  soil (if visible): {density: 1200, E: 5e5, nu: 0.30, material: "rigid"}

rubber_ducks_and_toys:
  toy: {density: [80, 150], E: [3e4, 5e4], nu: [0.4, 0.45], material: "elastic"}

sport_balls:
  ball: {density: [80, 150], E: [3e4, 5e4], nu: [0.4, 0.45], material: "elastic"}

soda_cans:
  can: {density: [2600, 2800], E: [5e10, 8e10], nu: [0.25, 0.35], material: "
metal"}

metal_crates:
  crate: {density: [2500, 2900], E: [8e7, 1.2e8], nu: [0.25, 0.35], material: "
metal"}

sand:
  sand: {density: [1800, 2200], E: [4e7, 6e7], nu: [0.25, 0.35], material: "sand"}

jello_block:
  jello: {density: [40, 60], E: [800, 1200], nu: [0.25, 0.35], material: "elastic
"}

snow_and_mud:
  snow_and_mud: {density: [2000, 3000], E: [8e4, 1.2e5], nu: [0.15, 0.25],
material: "snow"}
```

Figure 12: **In-Context Physics Condensed Examples.** Material properties for each object class used in the VLM prompting. Density is in kg/m³, E (Young's Modulus) is in Pa, nu (Poisson's ratio) is dimensionless.

```
We are trying to label a 3D object with physical properties. The physical
properties are:
    — Density
    — Young's Modulus
    — Poisson's Ratio
    — Material model

where the material model is one of the following: \{material_list_str\}
We have an automatic semantic segmentation model that can segment the object into
 different parts. We'll assume that each part has the same material model.

Your job is to come up with the part query to pass to the semantic segmentation
model, and the associated material properties for each part.
    \{special_notes\}
For example, for a \{class_name_for_example\}, the return is

    ```json
    \{example_material_dict_str\}
    ```
    \{example_explanation\}
Note that there are many different valid values for the material properties
including E, nu, and density that would influence how the object behaves. Thus,
instead of actual values, you should return a range of values like "E": [2e4, 2e6
]. Also, provide reasoning and constraints on the values when appropriate.

So the output should be a json with the following format:

    ```json
    \{\{
        "material\_dict": \{\{ ... similar to example\_dict with ranges ... \}\},
        "reasoning": "...",
        "constraints": "...",
        "all\_queries": "..."
    \}\}
    ```
Remember to write constraints in the form of python code. For example,
```python
    \{example_constraints_str\}
```
Note that you've been asked to generate a material range so 'material\_dict["
leaves"]["density"]' is a range of values. But for the purpose of the constraints
 writing, you can assume that the material\_dict["leaves"]["density"] is a single
 value, and generate the python code similar to the example above. This is
important because we will first sample a value from the range, then invoke your
constraints code. So instead of writing something like
```python
    assert material\_dict["leaves"]["density"][0] ...
```
you must write something like
```python
    assert material\_dict["leaves"]["density"] ...
```
Note that the correct code doesn't have the bracket because 'material\_dict["
leaves"]["density"]' will be already reduced to a single value by our sampler.
You will be provided with images of the object from different views or a single
view. Please try your best to come up with appropriate part queries as well. For
example, if the object doesn't have visible trunk or pot, then you should NOT
include them in the material\_dict. Only segment parts that are visible in the
image.
Also, because our CLIP segmentation model is not perfect, you should come up with
 alternative queries as well including the original queries in the all\_queries
list. For example,
    ```json
    \{example_all_queries_str\}
    ```
In total, you need to provide \{num_alternative_queries\} alternative queries.
Tips:
\{tips_str\}
— Make sure that each element in the 'all\_queries' list is in the exact same
order as the material\_dict keys.
                                        22
```

Figure 13: **VLM Actor System Prompt.**

```
You are a segmentation quality critic. Your task is to evaluate the quality of
segmentation results produced by a CLIP—based segmentation model.

You will be shown:
1. A set of original RGB images of a 3D object from different views
2. Segmentation results for different part queries

Your job is to:
1. Evaluate each segmentation query based on how well it separates the object
into meaningful parts
2. Score each query on a scale of 1—10 (10 being perfect)
3. Provide reasoning for your scores
4. Suggest improvements to the queries if needed

Consider the following factors in your evaluation:
— Does the segmentation properly separate the object into distinct, semantically
 meaningful parts?
— Are the boundaries of the segments accurate and clean?
— Is any important part of the object missed or incorrectly segmented?
— IMPORTANT: note that our imperfect CLIP segmentation model is heavily
dependent on the choice of part queries. Thus,
even if a query might not be semantically correct, as long as it is useful for
separating the object into distinct parts,
you should score it high.
— Bad queries would result in bad segmentation that are noisy or different parts
 are not correctly and/or clearly separated.

Your output should be a JSON in the following format:

‘‘‘json
{
  "query_evaluations": {
    "query_0": {
      "score": 8,
      "reasoning": "This query effectively separates the object into functionally
distinct parts. The boundaries are clean and consistent across different views."
    },
    "query_1": {
      "score": 3,
      "reasoning": "This query fails to distinguish important parts of the object,
 making it unsuitable for physical property assignment."
    },
    ...
  },
  "best_query": "query_1",
  "suggested_improvements": "Consider using more specific terms like 'ceramic pot'
 instead of just 'pot' to improve segmentation boundaries."
}
‘‘‘
where ‘query_{i}‘ is the i—th query in the "all_queries" list.

Be detailed in your reasoning and make concrete suggestions for improvements.
```

Figure 14: **VLM Critic System Prompt.** System instruction for evaluating segmentation quality and suggesting improvements.

Listing 1: In-context Physics Examples

```
{
    "tree": {
        "class_name_for_example": "ficus tree",
        "special_notes": "",
        "example_material_dict": {
            "pot": {"density": 400, "E": 2e8, "nu": 0.4, "material_id":
    get_material_id("rigid")},
            "trunk": {"density": 400, "E": 2e6, "nu": 0.4, "material_id":
     get_material_id("elastic")},
            "leaves": {"density": 200, "E": 2e4, "nu": 0.4, "material_id
    ": get_material_id("elastic")}
        },
        "example_explanation": textwrap.dedent("""
            For this, we assume that the pot is stationary, while the
    trunk and leaves are made of "elastic", which will make
            them sway in the wind. The stiffness (Young's Modulus) of the
     trunk is much higher than that of the leaves.
        """),
        "example_all_queries": [["leaves", "trunk", "pot"], ["green", "
    orange", "reddish-brown"]],
        "tips": [
            "In a scene, typically there's a stationary part that will
    serve to fix the object to the ground. Usually, it's the pot, or some
     base of the tree. You must set the material_id of the stationary
    part to 6. If there's no stationary part, then never mind.",
            "The higher the 'E' is, the stiffer the object is. E.g., so
    tree would sway less in the wind.",
        ]
    },
    "flowers": {
        "class_name_for_example": "flowers in a vase",
        "special_notes": "",
        "example_material_dict": {
            "vase": {"density": 500, "E": 1e6, "nu": 0.3, "material_id":
    get_material_id("rigid")},
            "flowers": {"density": 100, "E": 1e4, "nu": 0.4, "material_id
    ": get_material_id("elastic")}
        },
        "example_explanation": textwrap.dedent("""
            Here, the vase is designated as stationary (material_id=6),
    indicating it should not move or sway.
            The flowers are set to a more pliable or flexible material (
    like "elastic" = 0), so that they can sway
            if there's wind or slight motion. The stiffness (Young's
    Modulus) of the vase is much higher than that
            of the flowers, making the vase rigid and the flowers more
    flexible.
        """),
        "example_all_queries": [["vase", "flowers"], ["ceramic base", "
    petals"], ["blue vase", "pink flower"]],
        "tips": [
            "In a typical flower arrangement, the vase (or base) is
    stationary, so give that part material_id=6 if present.",
            "The higher the 'E', the stiffer the part. So the vase should
     have a higher E range than the flowers.",
        ]
    },
    "shrub": {
        "class_name_for_example": "typical three-part shrub",
        "special_notes": textwrap.dedent("""
        Dataset note: Shrubs in our dataset stand by themselves---there
    is no planter or base.
```

```
            You should therefore return only the shrub's structural parts
    and none of them are stationary.
        """),
        "example_material_dict": {
            "stems":    { "density": 300, "E": 1e5, "nu": 0.35, "
    material_id": get_material_id("elastic") },
            "twigs":    { "density": 250, "E": 6e4, "nu": 0.38, "
    material_id": get_material_id("elastic") },
            "foliage":  { "density": 150, "E": 2e4, "nu": 0.40, "
    material_id": get_material_id("elastic") }
        },
        "example_explanation": textwrap.dedent("""
            Return *ranges* instead of single values and accompany them
    with reasoning, Pythonic
            constraints, and alternative query lists.
        """),
        "example_all_queries": [
            ["stems", "twigs", "foliage"],
            ["woody stems", "thin branches", "leaves"],
            ["brown sticks", "small branches", "green leaves"]
        ],
        "tips": [
            "Provide exactly the parts visible (usually stems/twigs +
    foliage).",
            "1e4 <= E <= 1e6.",
            "Stems should be stiffest > twigs > foliage.",
            "No part uses material_id 6 because nothing is fixed to the
    ground.",
        ]
    },
    "grass": {
        "class_name_for_example": "",
        "special_notes": textwrap.dedent("""
            **Dataset note:** Grass patches are usually isolated;
    occasionally a visible soil patch is
            underneath. Include a "soil" part only if it is visible.
        """),
        "example_material_dict": {
            "blades": { "density": 80, "E": 1e4, "nu": 0.45, "material_id
    ": get_material_id("elastic") }
        },
        "example_explanation": textwrap.dedent("""
            Example A (typical isolated grass---no stationary part):
            ```json
            {
                "blades": { "density": 80, "E": 1e4, "nu": 0.45, "
    material_id": get_material_id("elastic") }
            }
            ```

            Example B (grass with visible soil):
            ```json
            {
                "soil":    { "density": 1200, "E": 5e5, "nu": 0.30, "
    material_id": get_material_id("rigid") },
                "blades": { "density":  80,  "E": 1e4, "nu": 0.45, "
    material_id": get_material_id("elastic") }
            }
            ```
            Return *ranges*, reasoning, constraints, and alternative
    query lists.
        """),
        "example_all_queries": [
          ["blades"],
          ["grass"],
```

```
            ["green stalks"]
        ],
        "tips": [
            "Segment only the visible parts (sometimes just \"blades\")
.",
            "If *no* soil visible:\nall_queries: [[\"blades\"],[\"grass
\"],[\"green stalks\"]]",
            "If soil *is* visible:\nall_queries: [[\"soil\", \"blades
\"],[\"dirt\", \"grass\"],[\"brown base\", \"green grass\"]]",
            "1e4 <= E <= 1e6.",
            "If soil present -> give it material_id 6 and ensure E_soil >
 E_blades.",
            "If soil absent -> no stationary part; material_id 6 should
not appear.",
        ]
    },
    "rubber_ducks_and_toys": {
        "class_name_for_example": "",
        "special_notes": textwrap.dedent("""
            IMPORTANT: For rubber ducks and toys, we want to treat the
entire object as a single part. Do not attempt to
            segment it into multiple parts. The object should be treated
as a single, bouncy rubber-like object.
        """),
        "example_material_dict": {
            "toy": {"density": [80, 150], "E": [3e4, 5e4], "nu": [0.4,
0.45], "material_id": get_material_id("elastic")}
        },
        "example_explanation": "",
        "example_all_queries": [["toy"], ["rubber toy"], ["yellow duck"],
 ["plastic toy"]],
        "tips": [
            "Always use material_id=0 (jelly) for bouncy rubber-like
behavior",
            "Keep E relatively low (around 1e3) for good bounce",
            "Density should be in the range of typical rubber/plastic
toys",
            "Poisson's ratio should be around 0.35 for rubber-like
behavior",
            "Make sure all queries in all_queries list are single-part
queries"
        ]
    },
    "sport_balls": {
        "class_name_for_example": "",
        "special_notes": textwrap.dedent("""
            IMPORTANT: For sport balls, we want to treat the entire ball
as a single part. Do not attempt to
            segment it into multiple parts (like surface patterns or
seams). The ball should be treated as a single,
            bouncy object.
        """),
        "example_material_dict": {
            "ball": {"density": [80, 150], "E": [3e4, 5e4], "nu": [0.4,
0.45], "material_id": get_material_id("elastic")}
        },
        "example_explanation": "",
        "example_all_queries": [["ball"], ["sport ball"], ["basketball"],
 ["round ball"]],
        "tips": [
            "Always use material_id=0 (jelly) for bouncy behavior",
            "Keep E relatively low (around 1e3) for good bounce",
            "Density should be in the range of typical sport balls",
            "Poisson's ratio should be around 0.35 for rubber-like
behavior",
```

```
1404              "Make sure all queries in all_queries list are single-part
1405       queries"
1406          ]
1407      },
1408      "soda_cans": {
1409          "class_name_for_example": "",
1410          "special_notes": textwrap.dedent("""
1411              IMPORTANT: For soda cans, we want to treat the entire can as
1412       a single part. Do not attempt to
1413              segment it into multiple parts (like the top, body, or label)
1414       . The can should be treated as a single,
1415              rigid metal object.
1416          """),
1417          "example_material_dict": {
1418              "can": {"density": [2600, 2800], "E": [5e10, 8e10], "nu":
1419       [0.25, 0.35], "material_id": get_material_id("metal")}
1420          },
1421          "example_explanation": "",
1422          "example_all_queries": [["can"], ["soda can"], ["aluminum can"],
1423       ["metal can"]],
1424          "tips": [
1425              "Always use material_id=1 (metal) for rigid metal behavior",
1426              "Keep E relatively high (around 1e8) for metal stiffness",
1427              "Density should be in the range of typical aluminum (around
1428       2700 kg/m^3)",
1429              "Poisson's ratio should be around 0.3 for metal behavior",
1430              "Make sure all queries in all_queries list are single-part
1431       queries"
1432          ]
1433      },
1434      "metal_crates": {
1435          "class_name_for_example": "",
1436          "special_notes": textwrap.dedent("""
1437              IMPORTANT: For metal crates, we want to treat the entire
1438       crate as a single part. Do not attempt to
1439              segment it into multiple parts (like the sides, top, or
1440       bottom). The crate should be treated as a single,
1441              rigid metal object.
1442          """),
1443          "example_material_dict": {
1444              "crate": {"density": [2500, 2900], "E": [8e7, 1.2e8], "nu":
1445       [0.25, 0.35], "material_id": get_material_id("metal")}
1446          },
1447          "example_explanation": "",
1448          "example_all_queries": [["crate"], ["metal crate"], ["metal box
1449       "], ["steel crate"]],
1450          "tips": [
1451              "Always use material_id=1 (metal) for rigid metal behavior",
1452              "Keep E relatively high (around 1e8) for metal stiffness",
1453              "Density should be in the range of typical metal (around 2700
1454        kg/m^3)",
1455              "Poisson's ratio should be around 0.3 for metal behavior",
1456              "Make sure all queries in all_queries list are single-part
1457       queries"
          ]
      },
      "sand": {
          "class_name_for_example": "",
          "special_notes": textwrap.dedent("""
              IMPORTANT: For sand objects, we want to treat the entire
       object as a single part. Do not attempt to
              segment it into multiple parts. The sand should be treated as
        a single, granular material.
          """),
          "example_material_dict": {
```

```
            "sand": {"density": [1800, 2200], "E": [4e7, 6e7], "nu":
    [0.25, 0.35], "material_id": get_material_id("sand")}
        },
        "example_explanation": "",
        "example_all_queries": [["sand"], ["sand pile"], ["sand mound"],
    ["granular material"]],
        "tips": [
            "Always use material_id=2 (sand) for granular behavior",
            "Keep E relatively high (around 5e7) for sand stiffness",
            "Density should be in the range of typical sand (around 2000
    kg/m^3)",
            "Poisson's ratio should be around 0.3 for sand behavior",
            "Make sure all queries in all_queries list are single-part
    queries"
        ]
    },
    "jello_block": {
        "class_name_for_example": "",
        "special_notes": textwrap.dedent("""
            IMPORTANT: For jello blocks, we want to treat the entire
    object as a single part. Do not attempt to
            segment it into multiple parts. The jello block should be
    treated as a single, soft, bouncy object.
        """),
        "example_material_dict": {
            "jello": {"density": [40, 60], "E": [800, 1200], "nu": [0.25,
     0.35], "material_id": get_material_id("elastic")}
        },
        "example_explanation": "",
        "example_all_queries": [["jello"], ["jello block"], ["gelatin"],
    ["bouncy block"]],
        "tips": [
            "Always use material_id=0 (jelly) for soft, bouncy behavior",
            "Keep E relatively low (around 1000) for good bounce and
    jiggle",
            "Density should be in the range of typical jello (around 50
    kg/m^3)",
            "Poisson's ratio should be around 0.3 for jello-like behavior
    ",
            "Make sure all queries in all_queries list are single-part
    queries"
        ]
    },
    "snow_and_mud": {
        "class_name_for_example": "",
        "special_notes": textwrap.dedent("""
            IMPORTANT: For combined snow & mud objects, we treat the
    entire mixture as a single deformable part.  Do **not**
            attempt to split it into separate snow and mud regions---the
    simulation will use one MPM material.
        """),
        "example_material_dict": {
            "snow_and_mud": {"density": [2000, 3000], "E": [8e4, 1.2e5],
    "nu": [0.15, 0.25], "material_id": get_material_id("snow")}
        },
        "example_explanation": "",
        "example_all_queries": [["snow and mud"], ["slush"], ["muddy snow
    "], ["wet snow"]],
        "tips": [
            "Always set material_id = 5 (snow) so the simulator uses the
    appropriate elasto-plastic snow model.",
            "Keep E around 1e5 (the config value) to match the intended
    softness.",
            "Density is markedly higher than fluffy snow because of the
    mud/water content---use roughly 2-3 g/cm^3 (2000-3000 kg/m^3).",
```

```
            "Make sure every list in 'all_queries' contains **one**
    phrase because this is a single-part object."
        ]
    },
}
```

## C THE EFFECTS OF HUMAN PRIOR ON PIXIEVERSE

PIXIEVERSE is labeled via VLMs using in-conext physics examples manually tuned by humans. A condensed version of these in-context examples is provided in Fig. 12 and the full prompt in Listing 1. These examples align the VLM's physical understanding with human's. In our ablation result, we found that removing these examples significantly results as shown in Tab. 2.

The main differences between PIXIEVERSE labeling and NeRF2Physics are

1. We use VLM to propose object-dependent segmentation while NeRF2Physics using LLM is essentially blind. Specifically, ur VLM actor proposes segmentation queries based on a set of mutli-view images of the object as shown in Fig. 11.
2. We use semantic proposals (e.g., "pot", "trunk") instead of material proposals (e.g., "leather", "stone") like NeRF2Physics did. Computing similarity directly between material name and CLIP features yields inaccurate and noisy segmentation as shown in Fig. 10. This also limits the generality of the NeRF2Physics since one material type (e.g., "elastic") can only have a fixed set of parameters in a scene. In contrast, PIXIE enables spatially-varying parameter specification: the leaves and the trunk of a tree while both belonging to the same "elastic" class can have vastly different young modulus, Poisson ratio and density as shown in Fig. 16.
3. We proposes multiple candidates (e.g., "pot,leaves" vs "base,folliage") and use a VLM critic to select the best based on CLIP-based segmentation while NeRF2Physics does not have any selection mechanism. Figure 10 show the dramatic segmentation quality across different queries, highlighting the need for multiple candidates and selection.
4. We also provide manually tuned in-context physics parameter examples.

These crucial differences contribute to much higher quality dataset labeling as shown in Tab. 2.

## D VLM AS A PHYSICS JUDGE

We utilize a VLM to evaluate the realism of different candidate videos. The videos are scored on the scale 1-5, and an optional reference video and the prompt describing the video (e.g,. "tree swaying in the wind") is provided. We also use Cotracker (Karaev et al., 2024) to annotate the videos with motion traces. The system prompt is provided in Fig. 15.

## E MODEL ARCHITECTURE

### E.1 OVERVIEW

We employ a 3D UNet-based architecture for both discrete material segmentation and continuous material parameter regression. The architecture consists of two main components: (1) a feature projector for dimensionality reduction, and (2) a 3D UNet backbone for spatial processing.

Table 2: **PIXIEVERSE Ablation.** The effect of in-context physics examples on data quality. We include the executionability rate, which computes the fraction of times that a physic simulation can be successfully run without numerical explosion, and the realism score judged by Gemini.

| Method | Exec. Rate ↑ | VLM Score ↑ |
|---|---|---|
| **W/ In-context Examples (Ours)** | 100.0% | $4.83 \pm 0.09$ |
| W/o In-context Examples | 62.5% | $1.34 \pm 0.30$ |
| NeRF2Physics (Zhai et al., 2024) | 45.0% | $1.09 \pm 0.28$ |

```
You are a physics—realism judge for animation videos.

You will be shown several candidate animations of the SAME 3D object responding
to the SAME textual prompt that describes its intended physical motion.

Your tasks:
1. Carefully watch each candidate animation.
2. Describe what's going on in the animation.
3. Evaluate how physically realistic the motion looks (0—5 scale).
4. Identify concrete pros / cons affecting the score (e.g. energy conservation
errors, temporal jitter, incorrect response to gravity, static etc.).
5. Suggest specific improvements.
6. Pick the overall best candidate.

Please output ONLY valid JSON with the following schema:
{
  "candidate_evaluations": {
    "candidate_0": {"description": str, "score": float, "pros": str, "cons": str,
 "suggested_improvements": str},
    "candidate_1": { ... },
    "candidate_2": { ... }
  },
  "best_candidate": "candidate_i", // the key of the best candidate
  "general_comments": str // any overall remarks (optional)
}

NOTE: ignore missing videos. Still return score for 'candidate_{idx}' that are
present.

NOTE: to make your job easier, we have also annotated the video with the Co—
Tracker. Cotracker is a motion tracker algorithm to highlight the moving parts in
 the videos.
Pay close attention to the motion traces annotated in the videos to gain
information on how the object is moving.
Note that for objects that barely move, there will still be dots in the Co—
Tracker video, but the motion
(lines) will be very short or non—existent, indicating that the points are not
moving.

Cotracker can sometimes produce noisy traces so only use it as a reference, and
consider the motion of the object as a whole, and other visual cues.
```

Figure 15: **VLM Evaluator's System Prompt.**

## E.2 FEATURE PROJECTOR

The feature projector is used when the input feature dimension differs from the conditioning dimension:

- **Input features**: The model supports three input modalities:
  - RGB features: $\mathbf{F} \in \mathbb{R}^{N \times 3 \times D \times H \times W}$
  - CLIP features: $\mathbf{F} \in \mathbb{R}^{N \times 768 \times D \times H \times W}$
  - Occupancy features: $\mathbf{F} \in \mathbb{R}^{N \times 1 \times D \times H \times W}$
- **Projection**: Features are projected to a unified conditioning dimension of 32 channels using a feature projector with hidden dimension of 128 (when input channels $> 32$). The projector consists of three layers of Conv3D, GroupNorm and SiLU activation.

## E.3 3D UNET ARCHITECTURE

We employ a U-Net architecture (Dhariwal and Nichol, 2021; Ronneberger et al., 2015) operating on 3D feature grids of shape $\mathbb{R}^{N \times 32 \times 64 \times 64 \times 64}$. The network follows a standard encoder-decoder structure with skip connections, using a base channel dimension of 64 and channel multipliers of [1, 1, 2, 4] across four resolution levels.

The encoder begins with a 3D convolution that projects the 32-dimensional input features to 64 channels. The encoder then processes features through four resolution levels, each containing three residual blocks. The first two levels maintain 64 channels while progressively reducing spatial dimensions from $64^3$ to $32^3$. The subsequent levels double the channel count at each downsampling step, reaching 128 channels at $16^3$ resolution and 256 channels at $8^3$ resolution. Downsampling between levels is performed using strided 3D convolutions with stride 2.

At the bottleneck, the network processes the lowest resolution features through a sequence of residual block, attention block, and another residual block, all operating at $8^3$ spatial resolution with 256 channels. Note that in our implementation, attention blocks are disabled by setting attention resolutions to empty.

The decoder symmetrically reverses the encoder path, utilizing skip connections from corresponding encoder levels. Upsampling is achieved through nearest-neighbor interpolation with a scale factor of 2, followed by 3D convolution. Each decoder level matches the channel dimensions and number of residual blocks of its corresponding encoder level.

Each residual block follows the formulation $\text{ResBlock}(x) = x + f(x)$, where $f$ consists of layer normalization, LeakyReLU activation with negative slope 0.02, 3D convolution with kernel size 3, another layer normalization and activation, dropout, and a final zero-initialized 3D convolution. When input and output channels differ, the skip connection employs a $1 \times 1 \times 1$ convolution for channel matching.

The final output layer applies layer normalization, LeakyReLU activation, and a 3D convolution that projects to either 8 channels for discrete material classification or 3 channels for continuous material parameter regression.

## F ADDITIONAL RESULTS

We visualize the physics predictions by our model in Fig. 16. Figure 17 breaks down the material accuracy across semantic classes of PIXIEVERSE between our PIXIE CLIP versus two ablated versions using RGB and occupancy input features. Figure 18 qualitatively compare the ablated methods on the real-world scenes. Table 3 includes the quantitative reconstruction results on real-world scenes using PSNR and SSIM metrics.

## G PIXIE AS INFORMED PRIOR FOR TEST-TIME OPTIMIZATION

Test-time optimization methods such as DreamPhysics and OmniPhysGS often rely on slow per-scene optimization to refine material parameters, and their performance is highly sensitive to initialization

Table 3: **Per-Category Reconstruction Performance.** We report PSNR and SSIM across three object categories (bun, burger, dog) and their mean from the Spring-Gauss (Zhong et al., 2024) dataset Higher is better.

| | bun | burger | dog | Mean |
|---|---|---|---|---|
| **PSNR ↑** | | | | |
| **CLIP (ours)** | **25.23** | **23.41** | **20.18** | **22.94** |
| RGB | 20.31 | 18.92 | 16.37 | 18.53 |
| Occupancy | 19.52 | 18.26 | 15.83 | 17.87 |
| OmniPhysGS | 19.50 | 18.21 | 15.72 | 17.81 |
| DreamPhysics | 20.79 | 19.28 | 16.60 | 18.89 |
| NeRF2Physics | 20.14 | 18.74 | 16.40 | 18.43 |
| **SSIM ↑** | | | | |
| **CLIP (ours)** | **0.920** | **0.875** | **0.852** | **0.882** |
| RGB | 0.866 | 0.821 | 0.797 | 0.828 |
| Occupancy | 0.871 | 0.826 | 0.804 | 0.834 |
| OmniPhysGS | 0.885 | 0.839 | 0.820 | 0.848 |
| DreamPhysics | 0.883 | 0.835 | 0.813 | 0.844 |
| NeRF2Physics | 0.889 | 0.842 | 0.823 | 0.851 |

(Sec. 4). Here we show that PIXIE's feed-forward predictions serve as *strong informed priors* for these methods. For the Ficus scene used in DreamPhysics, Fig. 19 compares optimization curves initialized either from DreamPhysics' default parameters or from PIXIE's predicted parameters. When initialized with PIXIE, the latent distillation loss (computed via ModelScoped video diffusion priors, as in (Huang et al., 2024)) converges faster and to a lower final value. This demonstrates that PIXIE can accelerate test-time optimization pipelines by providing a physically coherent, semantically consistent starting point for continuous parameters $(E, \nu, \rho)$ and material classes.

# H  GENERALIZATION TO REAL-WORLD OBJECTS: ABO-500 MASS ESTIMATION

To evaluate the robustness of the predicted density fields beyond synthetic data and NeRF-style real scenes, we further test PIXIE on the ABO-500 dataset (Zhai et al., 2024), which contains 500 real products sourced from Amazon with ground-truth mass. This real-world dataset contains many out-of-distribution objects such as furniture that was not in our training set. We estimate object mass by integrating PIXIE's predicted density over the reconstructed object volume. We compared the absolute difference error (ADE), absolute log difference error (ALDE), absolute percentage error (APE), and min ratio error (MnRE) following NeRF2Physics. As shown in Table 5, PIXIE achieves lower error across ADE, ALDE, APE, and higher MnRE compared to NeRF2Physics, despite never observing these object categories during training. This demonstrates strong semantic generalization and the physical plausibility of PIXIE's continuous predictions, reinforcing its utility for downstream real-world applications.

# I  HUMAN USER STUDY

We conduct a blind user study containing 512 responses and 16 volunteers. The object and method ordering are randomly shuffled in each trial to mitigate bias. Each trial displayed four side-by-side simulations rendered using material parameters from different methods (PIXIE, DreamPhysics, NeRF2Physics, OmniPhysGS). Participants ranked the realism of the animations, with ties allowed . As summarized in Tab. 4, PIXIE achieves the best average rank (1.78, lower is better) and is chosen as the top method in 55% of trials—over 20% higher than the next-best method. These results align with both Gemini-VLM evaluations and the qualitative trends in Fig. 5–6, confirming that humans also perceive PIXIE's motion as the most physically plausible.

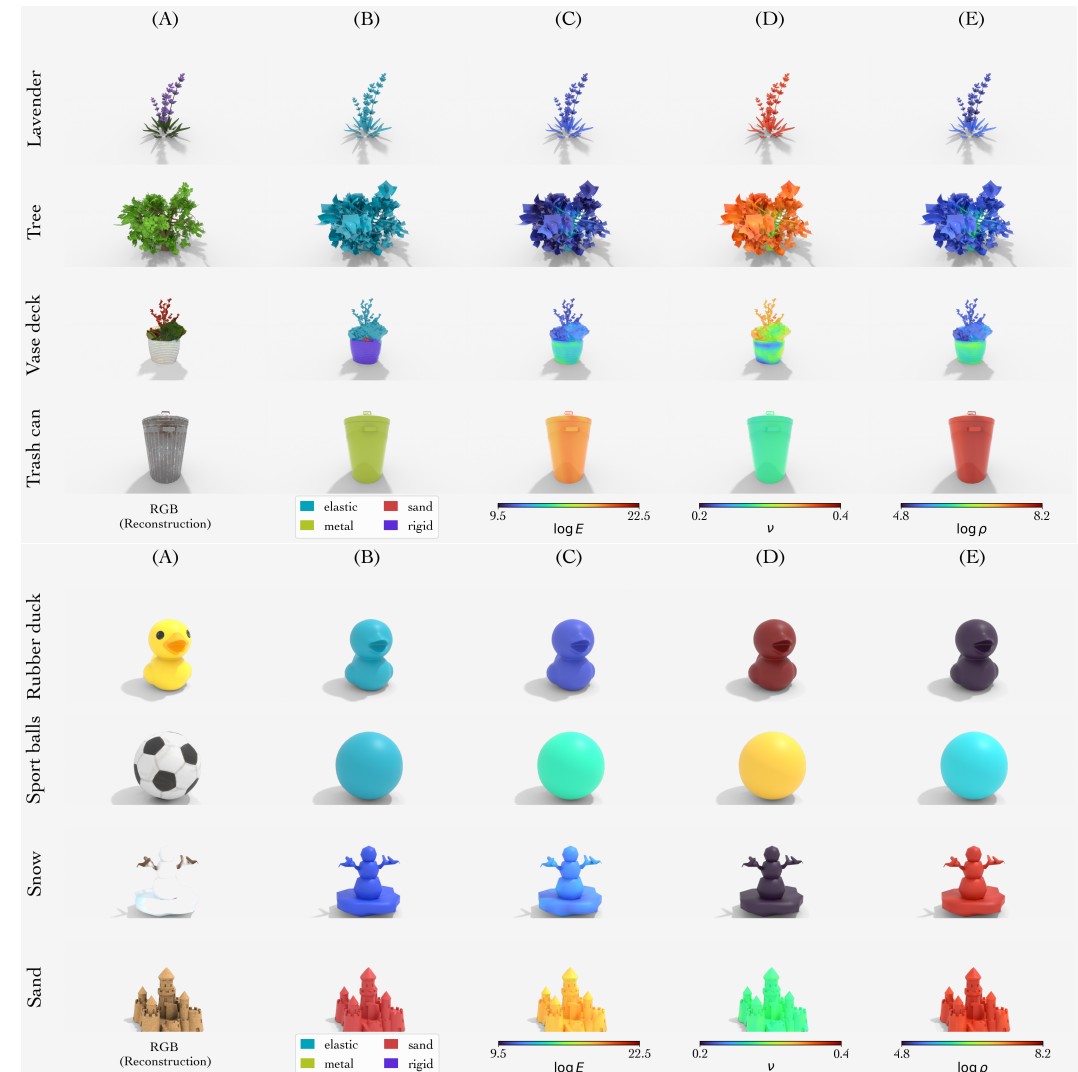

Figure 16: **PIXIE Prediction Visualization.** PIXIE simultaneously recovers discrete material class, continuous Young's modulus (E), Poisson's ratio ($\nu$), and mass density ($\rho$) with a high degree of accuracy. For example, the model correctly labels foliage as elastic and the metal can as rigid, while recovering realistic stiffness and density gradients within each object.

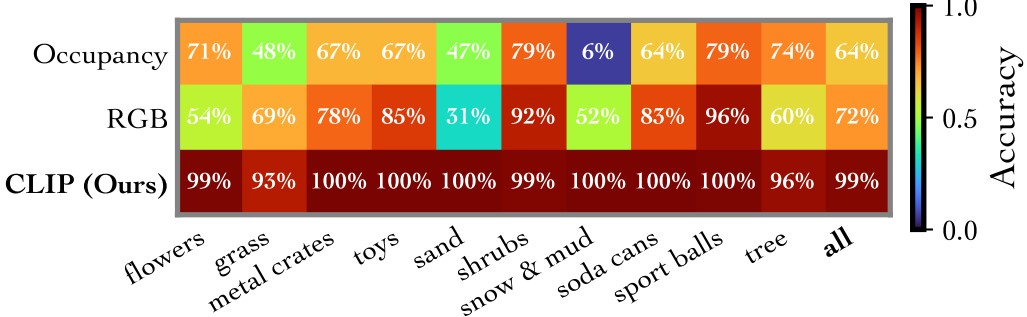

Figure 17: **PIXIE Ablation's Per-class Accuracy on synthetic scenes**. CLIP features generalizes in synthetic scenes, outperforming RGB and occupancy on all classes.

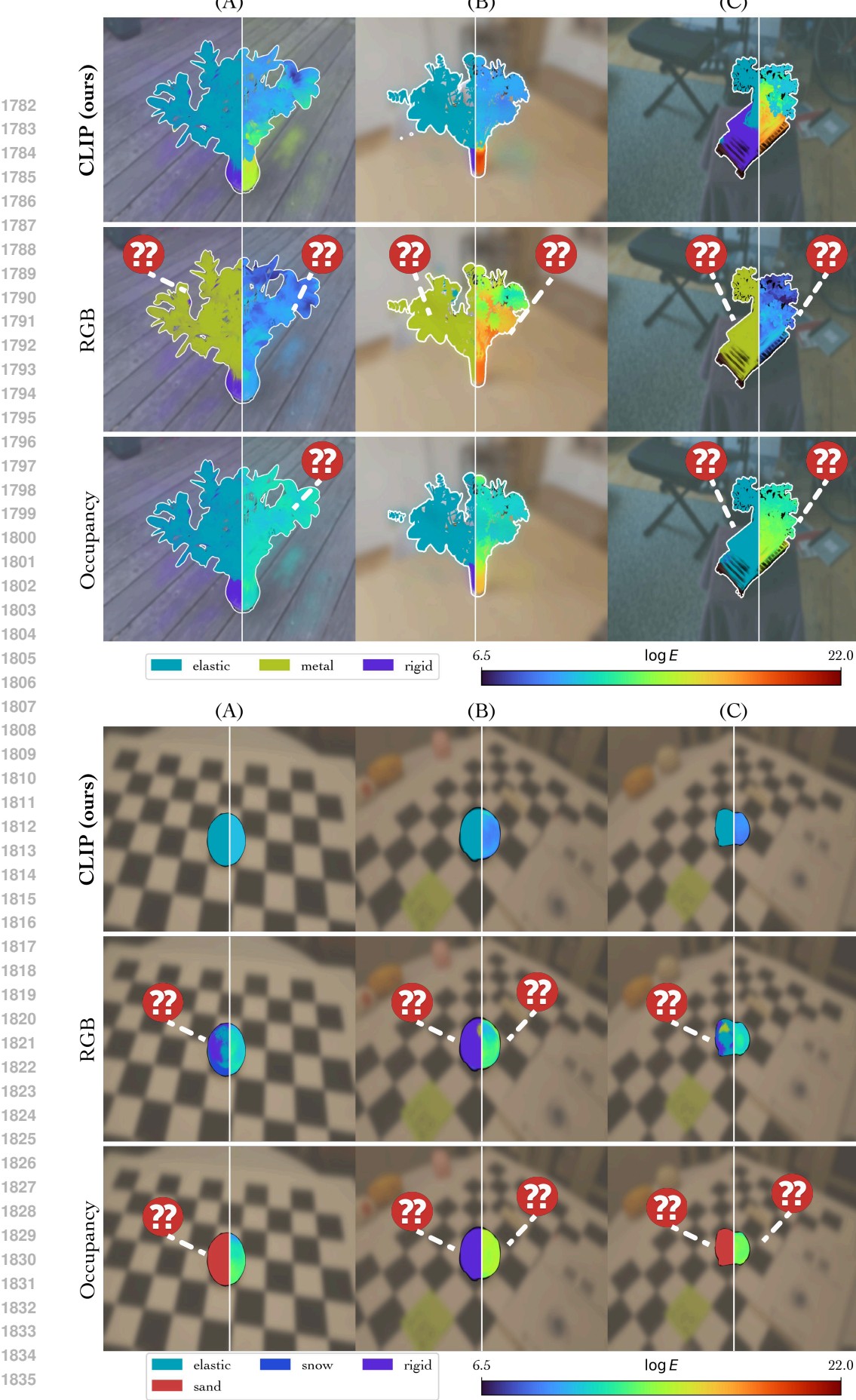

Figure 18: **Pixie's Feature Type Ablation on Real Scenes.** Replacing CLIP features with RGB or occupancy severely degrades the material prediction. Incorrect predictions such as leave mislaballed as metal or Young's modulus being uniform within an object are marked with question marks. This highlights the power of pretrained visual features in bridging the sim2real gap.

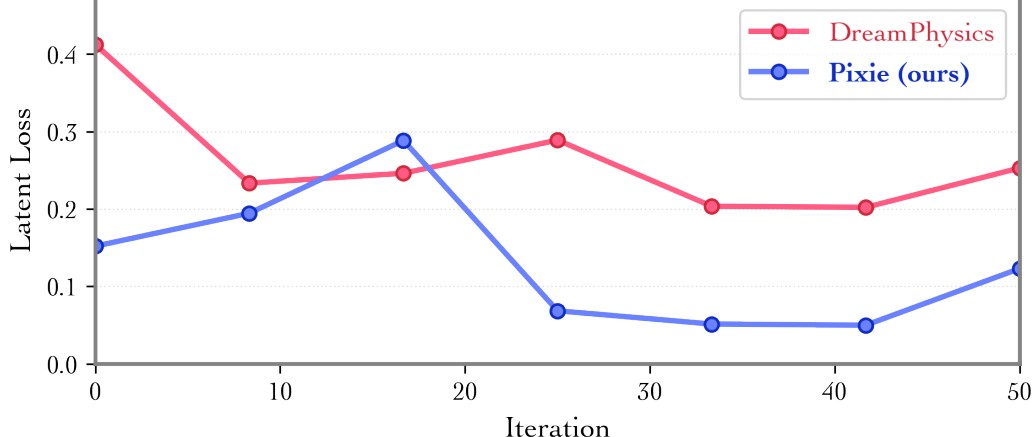

Figure 19: **PIXIE's predicted parameters can serve as an informed priors.** When initialized with PIXIE's predicted physic parameters, DreamPhysics (Huang et al., 2024) test-time optimization converge to a 2.05x lower latent loss compared to default initializations in the original work.

Table 4: **User Preference Study**. Results from our blind human study, where participants ranked the perceived physical realism of animations produced by each method. We report the mean rank (lower is better) and the proportion of trials in which a method was chosen as most realistic (higher is better). Based on 512 responses from 16 volunteers, these human judgments validate PIXIE's superior physical plausibility and align closely with the VLM-based evaluations presented in the main paper.

| Method | Avg Rank ↓ | Best % ↑ |
|---|---|---|
| **Pixie** | **1.7875** | **55.00%** |
| DreamPhysics | 3.7000 | 31.25% |
| NeRF2Physics | 2.3125 | 28.75% |
| OmniPhysGS | 2.1625 | 7.50% |

| Method | ADE ↓ | ALDE ↓ | APE ↓ | MnRE ↑ |
|---|---|---|---|---|
| NeRF2Physics | 8.730 | 0.771 | 1.061 | 0.552 |
| **Ours** | **8.231** | **0.654** | **0.875** | **0.584** |

Table 5: **Mass estimation results on the ABO-500 dataset (Zhai et al., 2024).** We estimate mass by integrating predicted density over object volume. Despite no training on these real object categories, PIXIE achieves lower error across all metrics compared to NeRF2Physics, demonstrating strong real-world generalization.

| Method | Preprocessing (s) | Inference (s) | Total Time (s) |
|---|---|---|---|
| DreamPhysics (mid) | 35.5 | 3811 | 3846.5 |
| OmniPhysGS (mid) | 35.5 | 10302 | 10337.5 |
| NeRF2Physics | 175.0 | 140.24 | 315.24 |
| **Pixie (Ours)** | **210.5** | **1.95** | **212.45** |

Table 6: **End-to-End Runtime Comparison (RTX A6000).** All methods require a 3D scene representation prior to physics prediction—DreamPhysics and OmniPhysGS rely on 3D Gaussian Splatting, NeRF2Physics uses a distilled NeRF, and PIXIE uses both. While Fig. 4 focuses on the core problem studied in this paper—*physics prediction*— here we report full end-to-end wall-clock time including preprocessing. On average, learning a 3DGS model requires 35.5 s (1,000 iterations) and distilled NeRF training requires 2.92 min (5,000 steps) on a single NVIDIA RTX A6000 GPU. Even with these preprocessing stages included, PIXIE achieves a total runtime of only 212.45 s (≈3.5 min), making it 1.48× faster than NeRF2Physics (315.24 s) and *orders of magnitude* faster than test-time optimization methods such as DreamPhysics and OmniPhysGS, which require hours to converge.

