# OpenReview forum: "Pixie: Fast and Generalizable Supervised Learning of 3D Physics from Pixels"
_ICLR.cc/2026/Conference — Submitted to ICLR 2026_

### Official Review · Reviewer_nTtB · 2025-10-20

**Soundness:** 3
**Presentation:** 3
**Contribution:** 2
**Rating:** 4
**Confidence:** 3

**Summary:**

The paper proposes PIXIE, a generalizable feed-forward neural network that predicts physical properties from 3D visual features. The authors further introduce a semi-automatic data annotation pipeline and release PIXIEVERSE, a large-scale dataset with physical material annotations. Experimental results show that PIXIE achieves a 1.46–4.39× improvement in realism scores and faster inference than prior test-time optimization approaches.

**Strengths:**

1. The paper is clearly written and easy to follow.
2. Compared with prior test-time optimization methods, PIXIE is a generalizable feed-forward architecture that achieves higher realism scores while being substantially faster at inference.
3. The authors introduce a semi-automatic data-annotation pipeline and release PIXIEVERSE, an open-source dataset of 3D objects with physical material annotations.
4. Despite being trained on synthetic data, PIXIE generalizes effectively to real-world data by leveraging pretrained visual features (e.g., CLIP).

**Weaknesses:**

1. Instead of NeRF + CLIP + voxelization followed by transfer to 3DGS, why not learn 3DGS directly with distilled CLIP features, which could be more direct and efficient?
2. Compared to previous methods such as DreamPhysics and OmniPhysGS, PIXIE relies on an auxiliary NeRF with distilled CLIP features, the acquisition of which can be time-consuming. It would be preferable to report the additional runtime overheads, including NeRF training, feature-field voxelization, and transfer to 3DGS.
3. The authors employ a feature projector to map CLIP features onto a low-dimensional manifold, whereas in NeRF, the original high-dimensional features are retained, which may lead to inefficiencies in runtime and memory usage. Previous works have addressed this issue by either utilizing an autoencoder or PCA to compress CLIP features into a lower-dimensional space [1,2], or by introducing a learnable linear layer that maps low-dimensional 2D renderings back to high-dimensional feature maps [3]. As these approaches learn low-dimensional representations directly in 3D space, they could potentially offer greater efficiency than the design adopted by the authors.
4. Since the PIXIEVERSE dataset contains only 10 semantic classes, it is unclear how well the model generalizes to unseen categories in real-world scenarios. This limitation may result in implausible physical parameter estimations or unstable visual outputs when encountering out-of-distribution objects.

[1] Qin, Minghan, et al. "Langsplat: 3d language gaussian splatting." Proceedings of the IEEE/CVF Conference on Computer Vision and Pattern Recognition. 2024.

[2] Yang, Jiawei, et al. "Emernerf: Emergent spatial-temporal scene decomposition via self-supervision." arXiv preprint arXiv:2311.02077 (2023).

[3] Zhao, Yanpeng, et al. "Dynamic Scene Understanding Through Object-Centric Voxelization and Neural Rendering." IEEE Transactions on Pattern Analysis and Machine Intelligence (2025).

**Questions:**

See weakness.

---

> ### Author Response · Authors · 2025-11-24
>
> Thank you for your review! Below we address your concerns and will incorporate your feedback into the paper. The revised paper includes the new results and text in blue.
>
>
> ### Why not learning 3DGS + CLIP directly?
>
> Thank you for the excellent question!
>
> - We have experimented with distilling the CLIP features into a 3DGS via Feature Splatting [Qiu et al] [L068] and found the downstream segmentation quality to be poorer than distilled CLIP. We attribute this to the difference between explicit (3DGS) vs implicit (NeRF) representation as the work [Qiu et al] acknowledged i.e., NeRF-based CLIP distillation is better than 3DGS-based due to the continuous representation of NeRF. We will include in the paper a qualitative comparison between CLIP quality of feature splatting and our NeRF-based field.
>
> - Using 3DGS instead of regular feature voxel grid would also necessitate the use of a complicated architecture such as a transformer to deal with the variable number of particles between scenes. As the first work in feed-forward prediction, we prefer to keep the method as simple as possible with a U-net architecture to demonstrate its effectivenes.
>
>
> ### The reported inference time does not take into account NeRF training, CLIP distillation, and 3DGS reconstruction.
>
> Our paper focuses on the problem of physics prediction. Therefore, Fig 4 reported the time (L105) taken for physics prediction. The problem of learning appearance and geometry is orthogonal but not the focus of the paper. We note that all methods require a 3D representation: namely, PhysDreamer and OmniPhysGS requires 3DGS, Nerf2Physics requires distilled NeRF while our method requires both. Nonetheless, following the suggestion, we include the end-to-end time statistics below. 3DGS learning takes 35.5 seconds on average (1,000 iterations) while distilled NeRF learning takes 2.9167 minutes (5,000 steps). All measured on a single Nvidia RTX A6000 GPU. Even with the preprocessing time included, Pixie (~212.5s or 3.5 minutes) is still 1.483 times faster than NeRF2Physics (315.24s) and order of magnitude faster than test-time optimization methods (DreamPhysics, OmniPhysGS), which take hours to complete.
>
>
> | **Method**        | **Preprocessing**                      | **Inference** | **Total Time (s)** |
> |-------------------|-----------------------------------------|---------------|---------------------|
> | DreamPhysics (mid) | 3DGS (35.5 s)                          | 3811 s    | 3846.5 s        |
> | OmniPhysGS (mid)   | 3DGS (35.5 s)                          | 10302 s   | 10337.5 s       |
> | NeRF2Physics       | Distilled NeRF (175.0 s)               | 140.24 s  | 315.24 s        |
> | Pixie (Ours)       | 3DGS (35.5 s) + NeRF (175.0 s) = 210.5 s | 1.95 s    | **212.45 s**        |
>
> We have included this Table in the paper (Tab. 6)
>
> ### Would projecting CLIP onto a lower dimension during rasterization be more efficient?
>
> We note that the CLIP-NeRF distillation is implemented highly efficiently at cuda kernels, only taking around 3 minutes in total to reconstruct both appearance and CLIP semantic features. Thus, it is preferable not to compress the semantic features too prematurely.
>
> Instead, we use a feature projector (L212, Appendix E) during the physics learning which is trained jointly and end-to-end with the UNet. This means that the projection is tightly coupled with the physics learning task, compressing information necessary for physics prediction.
>
> We will also add your references [1-3] to the manuscript.
>
> ### PixieVerse only contains 10 semantic classes, how can it generalize to unseen categories?
>
> 1. First, we note that our categories are very **broad**. For instance, the "rubber_ducks_and_toys" category contains all virtually all deformable household toy objects (Fig 7). With these categories, we were able to generate all dynamics provided by the MPM simulator that we built upon from PhysGaussian, and other prior works including OmniPhysGS, DreamPhysics.
> 2. All of our evaluations are conducted on a a hold-out test set (L293-295). This means that the results are reported on only unseen objects.
>
> We believe that this approach can be scaled up to more categories in a industrial setting. At the current academic scale, we already observe zero-shot generalization to real-world scenes despite the significant visual gaps between real and synthetic scenes.
>
> [Qiu et al] Feature Splatting: Language-Driven Physics-Based Scene Synthesis and Editing. 2024

---

> > ### Author Response · Authors · 2025-11-27
> >
> > Dear Reviewer nTtB,
> >
> > Thank you again for your review! As we near the end of the discussion period, we kindly ask that you review our response to your feedback and consider adjusting your score taking into account our revision.
> >
> > Please also do not hesitate to let us know if there is any more question or issue that we can address.
> >
> > Thank you.

---

> > > ### Comment · Reviewer_nTtB · 2025-11-28
> > >
> > > I thank the authors for providing a detailed rebuttal; most of my concerns have been adequately addressed.
> > >
> > > Regarding the generalization ability of Pixie, although the authors claim that the categories are very broad, it would be better to quantitatively support this argument through an ablation study. In particular, evaluating performance on out-of-distribution categories (not just on unseen objects).
> > >
> > > On the other hand, I agree with the reviewer 74dK that directly predicting physical parameters from CLIP features has inherent limitations. Exploring approaches that infer physical parameters conditioned on videos (either generated by video diffusion models or from realistic observations), or even directly from textual descriptions (e.g., “a hard box,” “a deflated ball”), would be an interesting direction for future work.

---

> > > > ### Author Response · Authors · 2025-11-28
> > > >
> > > > Dear Reviewer nTtB,
> > > >
> > > > Thank you for the feedback! Below, we address your additional points.
> > > >
> > > > ### Generality on unseen categories
> > > >
> > > > As you suggested, below we include some quantitative results on the ABO-500 dataset from NeRF2Physics. This dataset includes 500 real-world objects sourced from products sold on Amazon. The dataset contains many objects that are from **unseen categories** such as chair, stool, table, bookshelves. We compared the absolute difference error (ADE), absolute log difference error (ALDE), absolute percentage error (APE), and min ratio error (MnRE) of mass prediction following NeRF2Physics.
> > > >
> > > > | **Method**      | **ADE ↓** | **ALDE ↓** | **APE ↓** | **MnRE ↑** |
> > > > |-----------------|-----------|------------|-----------|------------|
> > > > | NeRF2Physics | 8.730 | 0.771  | 1.061 | 0.552  |
> > > > | Ours            | **8.231** | **0.654**  | **0.875**     | **0.584**  |
> > > >
> > > > ### Ground-truth video-based or video-model-based prediction
> > > >
> > > > We'd argue that there are two main challenges with current video-generation-based optimization methods:
> > > >
> > > > - Even SOTA video generation models currently struggle with generate accurate and consistent physics [1]. While the video generation model in works like PhysDreamer must generate the entire video taking appearance and physics into account, the job of the feed-forward model is much simpler: to generate a set of plausible parameters, which is then simulated by a reliable MPM solver.
> > > >
> > > > - Secondly, given the synthetic video, the video-based optimization is still very slow and noisy (Fig. 4), prone to local optima. Due to the highly non-linear optimization process, previous works including PhysDreamer, DreamPhysics, OmniPhysGS and so on can only estimate one quantity (e.g., Young's Modulus) while requiring the user to hand-tune other paramters (L304-305), defeating the purpose of automatic labeling. Our model and dataset annotation procedure, on the other hand, can inference all physics parameters (including Young's modulus, material class, Poisson's ratio, density) simulatenously (Fig 16).
> > > >
> > > > Regarding generating physic parameters from realistic observations, as our primary goal is construct interactive 3D virtual world, there is often no such ground-truths.
> > > >
> > > > We also agree that text-based physics generation is an interesting direction for future work. However, we note that our work is primarily focused on plausible physics prediction while text-based generation also requires generating the geometry and appearance, which is an orthogonal problem. We will add these future outlooks to the conclusion and future works Section.
> > > >
> > > > **Combining our method with video-based optimization**: Lastly, in the new results (Fig. 19) in the revised paper, we also showed that our feed-forward prediction can produce an informed prior for downstream video-based optimization method to further finetune and refine, leading to around 2.05x improvement in video-diffusion latent loss compared to default initializations.
> > > >
> > > >
> > > > Please feel free to let us know if there's any more analysis or experiments that can be run before the deadline that you'd like to see to improve your rating or confidence.
> > > >
> > > >
> > > >
> > > > [1] PhysCtrl: Generative Physics for Controllable and Physics-Grounded Video Generation, Wang et al 2025.

---

### Official Review · Reviewer_wPyp · 2025-10-28

**Soundness:** 3
**Presentation:** 3
**Contribution:** 3
**Rating:** 4
**Confidence:** 4

**Summary:**

The paper presents a supervised framework that predicts physical properties of 3D scenes directly from visual inputs. The method, PIXIE, trains a feed-forward 3D U-Net on CLIP-distilled volumetric features to infer both discrete material types and continuous physical parameters from multi-view RGB images. These predictions can be coupled with Gaussian Splatting and simulated using the Material Point Method (MPM) to produce realistic physics-based animations. To support training, the authors introduce PIXIEVERSE, a dataset of 3D assets labeled with physical material annotations across 10 semantic categories. Experiments show that PIXIE achieves higher realism scores and is three orders of magnitude faster than test-time optimization baselines, while also generalizing zero-shot to real-world scenes despite being trained solely on synthetic data.

**Strengths:**

- The paper introduces PIXIEVERSE, an open-source dataset of 1,624 3D assets annotated with physical material parameters, enabling future research.
- The paper proposes the first supervised learning method that directly predicts both discrete material classes and continuous physical parameters (Young’s modulus, Poisson’s ratio, density) from 3D visual features, which enables faster inference than prior test-time optimization methods.
- Extensive experiments on synthetic data and real-world data show the superior performance of the proposed method.

**Weaknesses:**

- The PIXIEVERSE dataset relies heavily on semi-automatic annotations generated by vision-language models. Such labels may contain systematic biases or noise, and their accuracy is not quantitatively validated.
- While Figure 4 reports a 2-second inference time, this does not account for the required NeRF/feature-field reconstruction step, which can be computationally expensive.
- Although the paper claims zero-shot generalization on Spring-Gaus data, it omits comparisons against Spring-Gaus, instead asserting that "no other baseline can generalize under this setting".
- Missing related work: It would be better if the author could compare with [1], which aims to estimate physical properties implicitly from videos.

[1] Zhu X, Deng H, Yuan H, et al. Latent Intuitive Physics: Learning to Transfer Hidden Physics from A 3D Video. ICLR 2024.

**Questions:**

1. In real-world environments with cluttered backgrounds and potentially moving objects, how does the proposed method identify and isolate the dynamic regions relevant for physical simulation?
2. The paper states that each Gaussian in the Gaussian Splatting model is treated as an MPM particle (Sec. 3.1), but this mapping might be uneven, as splats are not uniformly distributed and often concentrate near visible surfaces. Could this cause inconsistencies in material distribution or simulation stability? Have any corrective strategies been applied to mitigate these issues?

---

> ### Author Response · Authors · 2025-11-24
>
> Thank you for your review! Below we address your concerns and will incorporate your feedback into the paper. The revised paper includes the new results and text in blue.
>
>
> ### Is there procedure to validate the dataset to avoid noise or systematic bias?
>
> In Appendix B and C, we detailed the data curation and quality control process extensively. For instance, Fig 10 provides a qualitative example on how different choices of query terms can effect the segmentation quality. Tab 2 provides a quantitative assessment on different design choices in data annotations. Our dataset achieves a 100% execution rate and a VLM score of 4.83, far better than other ablated versions (Tab 2).
>
> Specifically, to mitigate CLIP noise with different query terms (Fig 10), we propose semantic querying and multiple candidates with VLM critic (Sec B3-B4). We also discussed reducing potential VLM's error via in-context examples in Appendix C.
>
> ### The reported inference time does not take into account reconstruction time
>
> Our paper focuses on the problem of physics prediction. Therefore, Fig 4 reported the time (L105) taken for physics prediction. The problem of learning appearance and geometry is orthogonal but not the focus of the paper. We note that all methods require a 3D representation: namely, PhysDreamer and OmniPhysGS requires 3DGS, Nerf2Physics requires distilled NeRF while our method requires both. Nonetheless, following the suggestion, we include the end-to-end time statistics below. 3DGS learning takes 35.5 seconds on average (1,000 iterations) while distilled NeRF learning takes 2.9167 minutes (5,000 steps). All measured on a single Nvidia RTX A6000 GPU. Even with the preprocessing time included, Pixie (~212.5s or 3.5 minutes) is still 1.483 times faster than NeRF2Physics (315.24s) and order of magnitude faster than test-time optimization methods (DreamPhysics, OmniPhysGS), which take hours to complete.
>
>
> | **Method**        | **Preprocessing**                      | **Inference** | **Total Time (s)** |
> |-------------------|-----------------------------------------|---------------|---------------------|
> | DreamPhysics (mid) | 3DGS (35.5 s)                          | 3811 s    | 3846.5 s        |
> | OmniPhysGS (mid)   | 3DGS (35.5 s)                          | 10302 s   | 10337.5 s       |
> | NeRF2Physics       | Distilled NeRF (175.0 s)               | 140.24 s  | 315.24 s        |
> | Pixie (Ours)       | 3DGS (35.5 s) + NeRF (175.0 s) = 210.5 s | 1.95 s    | **212.45 s**        |
>
> We have included this Table in the paper (Tab. 6)
>
> ### Comparison against Spring-Gauss
>
> As explained in the Related Work (L139), Spring-Gauss is a test-time optimization method that iteratively optimizes the physic parameters using ground-truth videos. This method leverages Spring-Mass physics solver, which is mechanistically different from Eulerian-Langarian MPM we used. In order to limit cofounding factors and isolate the effects of the core technique (feedfoward vs test-time optimization) vs physics solver, we compared against other test-time optimization methods done on MPM in the paper including DreamPhysics and OmniPhysGS.
>
> The assertion is against the methods compared in the paper as evidenced in (Fig. 18, Tab 3).
>
> ### Missing Related Work
>
> Thank you for the reference. This work [1] is also a test-time optimization, requiring multi-view 3D videos to optimize the latent code. The work is also limited to only fluid simulation while Pixie is feed-forward and is a general methods that works for other types of materials like snow, sand, metal, elastic, rigid. However, while other prior works (L137-150) optimizes spatially varying physical paramters, [1] proposes to estimate the latent code, which can ease and speed up the optimization process.
>
> We've included the paper in the related work [L139].
>
> ### How does the method work for cluttered envrionments?
>
> Depending on the types of forces and interactions, identifying and cropping "dynamic" regions might not be needed. For example, if the force is wind, we can simply apply wind to the entire scene without manual cropping. For interaction such as dropping an object to the floor, the object must be cropped. Currently, we are cropping these objects manually as prior works do. But segmentation models like SAM3D can also be used!
>
> ### How would MPM work with non-uniform 3DGS particles?
>
> Thank you for raising this excellent question! Indeed, vanilla 3DGS is typically optimized for optimal rendering results so the particle distribution might be unsuitable for physic simulators like MPM. Like other MPM-based works, we follow PhysGaussian to mitigate these issues by using techniques like internal fillings, anistropy regularizer, and incremental evolution of Gaussians. Please refer to Sec 3 of the PhysGaussian paper for more details.

---

> > ### Author Response · Authors · 2025-11-27
> >
> > Dear Reviewer wPyp,
> >
> > Thank you again for your review! As we near the end of the discussion period, we kindly ask that you review our response to your feedback and consider adjusting your score taking into account our revision.
> >
> > Please also do not hesitate to let us know if there is any more question or issue that we can address.
> >
> > Thank you.

---

### Official Review · Reviewer_74dK · 2025-10-30

**Soundness:** 2
**Presentation:** 3
**Contribution:** 2
**Rating:** 2
**Confidence:** 4

**Summary:**

This paper proposes a feed-forward model to directly predict material parameters for 3D objects, from an aligned and distilled CLIP field. In order to train the model, the paper also proposes a dataset with 3D objects and material annotation pairs, aligned by both VLM models and humans. The proposed model shows great efficiency and good results on synthetic dataset.

**Strengths:**

1. Learning physics is meaningful and important for visual understanding and embodied AIs.

2. The proposed method does not require test-time optimization, making it efficient in parameter estimation.

3. The paper is well-organized and very easy to follow.

**Weaknesses:**

Despite the strengths above, this paper has the following weaknesses:

1. The motivation is problematic. The material parameters are purely predicted based on static semantics. Although the paper claims this as an advantage, this is a significant weakness in my opinion.

Firstly, even for the same material, parameters can vary widely. For example, rubber can be either hard or soft.

Secondly, material properties and mass distribution are interdependent, as demonstrated in [1]. This means that identical materials with different mass distributions can exhibit different "equivalent" parameters. For instance, the iron in a box can be made thicker, resulting in a higher effective Young’s modulus.

Fundamentally, these properties cannot be inferred from static semantic information alone. Therefore, the proposed method faces a theoretical limitation that cannot be overcome simply by scaling up or modifying the model under the current assumptions.

2. Following weakness 1, the method lacks flexibility to adjust its predictions when the semantic-material relationship falls outside the training distribution. If the initial prediction is incorrect, additional observations will not correct it. For example, if the training set only includes hard rubber, the model will consistently predict rubber as hard, even when it appears soft in real-world videos.

3. Baselines as Vid2Sim should be compared to demonstrate the basic assumption that only static images are needed.

[1] Takuhiro Kaneko, Structure from Collision, CVPR 2025

**Questions:**

1. How to do define material type $\ell$ in the proposed model?

2. Since different materials need different parameters, how to decide which set of parameters to learn for different kinds of objects?

---

> ### Author Response · Authors · 2025-11-24
>
> Thank you for your review! Below we address your concerns and will incorporate your feedback into the paper. The revised paper includes the new results and text in blue.
>
> ### Why does the problem formulation make sense?
>
> While we acknowledged in the Limitation Section (L428) that there is ambiguities in prediction, we'd like to argue why our formulation still makes sense and is valuable. It comes down to the distinction between **generation** and **reconstruction** tasks. While previous works seek to reconstruct virtual parameters from ground-truth videos, we focus on **generating** plausible physics parameters from static scenes. The literature on generation tasks in computer vision and robotics is vast (e.g., see [1, 2] for a survey) and there are many successful commercial applications using generative AI such as StableDiffusion, Veo3, SAM3D to name a few. In generation, there is inherent ambiguity. For instance, if we ask Veo3/Sora or text2image model to generate "a picture/video of a cat playing piano", it can still produce some useful outputs in the face of ambiguities (how big is the cat, what breed, what type of piano ect). Therefore, generation models like ours can still be very useful for applications in robotics [2], graphics and computer vision [1].
>
> ### The method lacks the flexibility to adjust to additional information when such information is present?
>
> - First, the dataset contains a diverse range of dynamics (Appendix B). Each object is labeled with a range of parameters (L748), and we use rejection sampling to construct the training dataset as detailed in L747-754, L946-952. This ensures that the dataset contains a wide range of dynamics (from soft to stiff balls, soft to stiff trees and so on). Thus, to put it in the context of your example, our dataset does contain both soft and hard rubber, and will thus not only produce hard rubber.
> - Secondly, the prediction provided by Pixie can still be used as an informed prior for downstream test-time optimization approaches. Recall that prior test-time optimization methods (L137-150) work by iteratively updating the physics parameters using the prior from a video diffusion model or reconstruction loss against ground-truth videos. However, as discussed in the paper, due to the highly nonlinear dynamics, most methods can only optimize a subset of parameters. DreamPhysics only optimize Young Modulus (L304) while other parameters need to be hand-tuned. Even for the optimized quantity, the initialization needs to be carefully tuned to be not too far from the plausible range. With Pixie, we can initialize all parameters to reasonable quantities which can be further fine-tuned by other test-time optimization.
>
> In Fig 19 of the updated manuscript, we show that using Pixie's predicted parameters as priors lead to much faster convergence and lower loss compared to default initialization.
>
> In this case, Pixie can complement other baselines.
>
> ### Why wasn't Pixie compared against Vid2Sim?
>
> Vid2Sim (L146-150) is a test-time optimization method that iteratively optimizes the physic parameters using ground-truth videos. This method leverages Linear Blend Skinning (LBS) pioneered by Simplicits [3], which is mechanistically different from Eulerian-Langarian MPM we used. In order to limit cofounding factors and isolate the effects of the core technique (feedfoward vs test-time optimization) vs physics solver, we compared against other test-time optimization methods done on MPM in the paper including DreamPhysics and OmniPhysGS.
>
>
> ### How to define the material type?
>
> As we explained in the method [L192-199], material type is the constitutive law in MPM method, specifying both the choices of hyperelastic energy function and return mapping. Please refer to appendix A3 for more details.
>
> ### Different types of materials need different parameters, how to decide which set of parameters to learn for different type of objects?
>
> As we explained in the method (L190-199) and the Appendix A, all objects require a material type (i.e., constitutive model), a Young's modulus, Poisson's ratio, and density. We simultaneously predict all those parameters (L328, Fig 16) while other prior works typically only choose to predict a subset (e.g., Young's modulus), requiring the user to hand-tune other parameters.
>
>
> [1] Generative Physical AI in Vision: A Survey, Liu et al 2025.
>
> [2] Survey on Modeling of Human-made Articulated Objects, Liu et al 2025.
>
> [3] Simplicits: Mesh-Free, Geometry-Agnostic, Elastic Simulation, Modi et al, 2024.

---

> > ### Comment · Reviewer_74dK · 2025-11-27
> > **Comments**
> >
> > Thanks for the authors’ response to address my concerns. However, my concerns are not addressed from the rebuttal either to me or to other reviewers.
> >
> > 1. I am respectfully against the point that this is a **generation** work in the authors’ response. The core goal of the paper is to accurately infer the physical properties of existing objects from 3D visual features. It is a regression task focused on "input-to-target mapping" rather than generating new. In other words, this is a *deterministic* mapping but not generation. And generalizing to new data domain is not generation either.
> >
> > 2. The authors misunderstood my two examples in weakness 1. What I am trying to express is that mapping from visual semantics to physical properties **staticly purely from images** is an ill-posed problem. Even if there are different kinds of rubber in your dataset, rubber with the same appearance can show extremely various properties in our daily life. This cannot be solved by simply inputting more data into the training data without changing the model assumption, because you will include many data pairs with the same input (visual semantics) but different output (physics parameters) and try to fit it with a deterministic function. What the current model can learn is only an **average** or **expected** material. This is the core weakness and my concern.
> >
> > 3. The authors misunderstood my question about material setting and parameters. I know what is constitutive law and how mpm works well. What I want to ask is for one clip feature, does the proposed method simultaneously predict all the parameters required by different materials and select the optimal one by the predicted material type?
> >
> > Based on the above weaknesses, it is hard to make a possitive recommendation.

---

> > > ### Author Response · Authors · 2025-11-27
> > >
> > > Dear Reviewer 74dK,
> > >
> > > Thank you for providing additional clarification and feedback regarding your concerns. Below, we respond to each of your points.
> > >
> > > ### Pixie is not a generation method because there is a deterministic mapping from input to target.
> > >
> > >  As advocated in the paper [L428], probabilistic models is a promising future direction already supported by our dataset [L747-754, L946-952]. Nonetheless, we'd argue that what makes a method "generative" as opposed to "reconstruction" in the previous response is not directly tied to whether a method is deterministic. In fact, many early works in generative AI have deterministic MLP architectures including Generative Adversarial Networks [2], novel-view synthesis such as NeRF [3] and Gaussian Splatting [4]. Follow-up works from 2020s including PixelNeRF [5], SparseNeRF [6], NeRFFusion [7] improves upon NeRF by generating a 3D scene from a handful or sometimes a single image, again purely using deterministic architectures. Only in later works such as VAE-NeRF [8] and  "Diffusion with Forward Models" [9] were the probabilistic architectures introduced.
> > >
> > > As own work focuses on generating plausible physical fields [L045, L113, L417] and is the "first step towards learning a supervised 3D model for physical material prediction" [L429], we chose a simple feed-forward architecture to highlight the power of learning against previous test-time optimization methods.
> > >
> > >
> > > ### Predicting physical properties from static images is ill-posed
> > >
> > > 1. __Predicting plausible physics from static scene is a studied problem of interests in previous works__: As surveyed in the related work [L131-136], many previous works including NeRF2Physics, PUGS, AutoVFX, FeatureSplatting, Physgen3d, Video2game, Physx produces physical properties of static scenes either via manual human assignment or LLM prediction. Our work also infers plausible material fields [L045] from static scene via supervised learning.
> > >
> > > 2. __Predicting the world properties from static and underspecified inputs is well-studied__: As we noted in our previous response, generation is inherently underspecified. Nonetheless, it is possible and often desirable to generate plausible predictions in the face of ambiguities. For instance, the reviewer can refer to our Veo3/Sora analogies or the earlier works in sparse-view and single-view NeRF predictions such as PixelNeRF.
> > >
> > > 3. __Human can also infer plausible physical properties from static scene__: Humans can also look at static scenes and have certain expectation about how the physics would behave [L041, L083-L085, L362]. For example, we can rely on the visual features of a tree and anticipate its movement under a gust of wind, or anticipate a vasedeck deformation without having to drop it on the floor, or a wall's stiffness without having to collide with it.
> > >
> > > 4. __Our method can produces informed priors for test-time optimization method__: As provided as a new result from our previous response (Fig. 19),  our method can provide a good initialization for all parameters to reasonable quantities which can be further fine-tuned by other test-time optimization (e.g., when other inputs such as multi-view ground-truth videos are available). So even if the model learns the __expected__ value of certain materials, the prediction can also be further improved when other dynamic inputs are available.
> > >
> > > We also appreciate the reviewer's point about "equivalent" parameters. The reviewer suggested that "the iron in a box can be made thicker, resulting in a higher effective Young's modulus". We agree that trading off different parameters such as mass density and young's modulus can produce indistinguishable behavior and generated videos. However, this is not a weakness of the proposed method due to the following reasons:
> > >
> > > 1. In many realistic settings, material parameters are only identifiable up to an equivalence class that preserves the observed motion. Our feed-forward model predicts one such plausible parameter set. Even other video-based test-time optimization alternatives can also converge to one prediction set at best. To measure the "true" physics, one would need to conduct complex tests such as tensile, strain, fluid displacement in a lab setting, often using heavy machinery such as the UTM.
> > > 2. As the goal of our paper and many others' previous works is to create "interactive and realistic virtual worlds" [L40, L113], it is enough to predict physics up to an equivalence class. In fact, what presented to the end-user is not the predicted physic quantities but rather than generated videos or object interactions resulting from such predictions.

---

> > > > ### Author Response · Authors · 2025-11-27
> > > >
> > > > ### Does the proposed method simultaneously predict all the parameters required by different materials and select the optimal one by the predicted material type?
> > > >
> > > > The method does simulatenously predict all physics parameters [L328, L1761, L257, L232]. We're also happy to provide additional clarification, should the reviewer can elaborate their question!
> > > >
> > > >
> > > >
> > > > [2] I. Goodfellow, J. Pouget-Abadie, M. Mirza, B. Xu, D. Warde-Farley,
> > > > S. Ozair, A. Courville, and Y. Bengio, “Generative adversarial networks,” Communications of the ACM, vol. 63, no. 11, pp. 139–144,
> > > > 2020
> > > >
> > > > [3] K. Gao, Y. Gao, H. He, D. Lu, L. Xu, and J. Li, “Nerf: Neural radiance field in 3d vision, a comprehensive review,” arXiv preprint arXiv:2210.00379, 2022.
> > > >
> > > > [4] B. Fei, J. Xu, R. Zhang, Q. Zhou, W. Yang, and Y. He, “3d gaussian splatting as new era: A survey,” IEEE Transactions on Visualization and Computer Graphics, 2024
> > > >
> > > > [5] pixelNeRF: Neural Radiance Fields from One or Few Images, Yu et al, 2020
> > > >
> > > > [6] SparseNeRF: Distilling Depth Ranking for Few-shot Novel View Synthesis, Wang et al, 2023
> > > >
> > > > [7] NeRFusion: Fusing Radiance Fields for Large-Scale Scene Reconstruction, Zhang et al, 20222
> > > >
> > > > [8] NeRF-VAE: A Geometry Aware 3D Scene Generative Model, Kosiorek et al, 2021
> > > >
> > > > [9] Diffusion with Forward Models: Solving Stochastic Inverse Problems Without Direct Supervision, Tewari et al, 2023

---

### Official Review · Reviewer_dw3d · 2025-11-01

**Soundness:** 3
**Presentation:** 4
**Contribution:** 2
**Rating:** 6
**Confidence:** 5

**Summary:**

This paper presents a novel approach to predict physical parameters of a dynamic object to be used in MPM-based simulation. This is done by curating a large-scale paired synthetic dataset of dense annotations of physical parameters of dynamic objects, and training a feed-forward amortized inference model to predict the physical parameters, only considering visual input.

In short, we can consider this paper to be doing the following: it uses a VLM-based semi-automated pipeline to extract physical parameters, and uses a neural network to perform amortized inference. So basically, the model is trying to memorize how the semi-automated pipeline works by learning on some synthetic data. Therefore, I think the core contribution of this paper should be the data curation framework.

**Strengths:**

- The paper is well-written, with a very clear flow of the story. The figures, as well as the supplementary website, are aesthetically pleasing and convey the results pretty well.
- This paper is one of the first works that attempt to perform physical parameter prediction of 3D objects in a feed-forward manner, and this indeed leads to a reduced time for test-time optimization.
- The qualitative results and renderings look very nice.

**Weaknesses:**

- The collected dataset only contains 10 categories of objects. This is partially because of the highly engineered design of their data annotation framework - it seems to be very complex and ad-hoc, and I do not believe that it can be really scaled up to model a more diverse range of object categories, not to mention that there are a lot of objects that simply cannot be categorized into some categories.
- There is no evaluation on the reliability of the data curation process. How reliable is the proposed framework? How hard is it to migrate to include more categories? An actionable plan would be collecting some real-world objects where you can measure the physical parameters, and compare the result produced by the data curation pipeline and the ground-truth physical parameter.
- The evaluation protocol, as described in L323, is vague. What does "manually verified by humans" mean? How large is the test dataset?
- Using VLM to judge a video's quality is not reliable. A user study is needed to show whether the generated video is really better than the baselines.
- How can the evaluation be based on reconstruction quality, given that the physical parameters are inherently ambiguous, given only a single image?
- Also, it's not fair to me to only compare on the proposed dataset, as the model is explicitly trained on this data, yet other methods do not have this information.
- It would be very helpful if there could be quantitative comparisons with baselines on the real-world demos.

**Questions:**

- How does the proposed model deal with the ambiguity of the physical parameter prediction? It now only takes multi-view images as input, and it is clearly not enough to determine the dynamic physical parameters. Will the neural network tend to produce averaged parameter sets? This leads to a larger concern about the scalability of the proposed pipeline - for a more complex dataset, there will be more ambiguities. How will this framework handle that?
- Is it possible to combine video model-based physical parameter estimation with the proposed pipeline?  For example, is it possible to use PhysDreamer to annotate parameters for the dataset?
- L274: unfinished sentence?

---

> ### Author Response · Authors · 2025-11-24
>
> Thank you for your review! Below we address your concerns and will incorporate your feedback into the paper. The revised paper includes the new results and text in blue.
>
> ### Core contribution of the paper.
> We'd like to re-iterate the core contributions as outlined in the paper and to offer some more perspective regarding the reviewer's summary of the paper. Our contributions are:
> - We introduces Pixie, "a unified framework that predicts discrete material types and continuous physical parameters" (L101, Fig 16) directly from visual features. In contrast, most prior works only predict a subset of parameters (L302-317).
> - Pixie is the first feed-forward model that can predicts MPM-ready physical parameters. (L46, 105), enabling extremely fast inference and accurate predictions.
> - The training of the model is not trivial either. For instance, naively training on just occupancy or RGB features leads to significantly worse results (Tab 1, Fig 18) while leveraging pretrained visual features such as CLIP leads to markedly better results (Tab 1) and even zero-shot generalization to real-world scenes (L108, Fig 6).
>
> PixieVerse was collected to train such a feed-forward model as because large-scaled 3D datasets such as Objaverse exists, they do not contained detailed spatially varying physics annotations. (L91-98)
>
> Additionally, we'd argue that the model learns from the dataset instead of "memorizing".
> - All of our evaluations are conducted on a a hold-out test set (L293-295). The model cannot memorize something it has not seen before.
> - The dataset does not contain any real-world dataset yet the model can still zero-shot generalize to such scenes from common datasets like NeRF, LERF (L295, Fig 6).
>
> ### The dataset contains 10 categories, how hard is it to scale to more categories?
> First, we note that our categories are very **broad**. For instance, the "rubber_ducks_and_toys" category contains all virtually all deformable household toy objects (Fig 7). With these categories, we were able to generate all dynamics provided by the MPM simulator that we built upon from PhysGaussian, and other prior works including OmniPhysGS, DreamPhysics.
>
> Secondly, most of the annotation process is automatic, making it very amenable to scaling to more categories. The only two steps that require a human are the optional manual correction for object filtering (Fig 9) and providing in-context examples (L1188). The manual correction takes around 1.32 minute per categories and tuning the physics config based off PhysGaussian default config takes around 5.8 minutes per category on average.
>
> ### Where is the evaluation of the reliability of the data curation and how reliable is this process?
>
> - In Appendix B and C, we detailed the data curation and quality control process extensively. For instance, Fig 10 provides a qualitative example on how different choices of query terms can effect the segmentation quality. Tab 2 provides a quantitative assessment on different design choices in data annotations. Our dataset achieves a 100% execution rate and a VLM score of 4.83, far better than other ablated versions (Tab 2).
> - As you suggested, below we include the mass estimation results on the ABO-500 dataset from NeRF2Physics. This dataset includes 500 real-world objects sourced from products sold on Amazon. We compared the absolute difference error (ADE), absolute log difference error (ALDE), absolute percentage error (APE), and min ratio error (MnRE) following NeRF2Physics. Our model predicts density, which is used for computing approximate mass by summing over known object volume.
>
> | **Method**      | **ADE ↓** | **ALDE ↓** | **APE ↓** | **MnRE ↑** |
> |-----------------|-----------|------------|-----------|------------|
> | NeRF2Physics | 8.730 | 0.771  | 1.061 | 0.552  |
> | Ours            | **8.231** | **0.654**  | **0.875**     | **0.584**  |
>
> ### What does "manually verified by humans" (L323) mean?
>
> The test-set contains 38 synthetic scenes (L293). A human looks at the generated dynamics and also predicted parameter visualization (Fig. 5) to validate the reasonableness of the realism score. For instance, in Fig 5, scene C by visualizing the predicted E, we can explain the stiffness of the results and low VLM scores, guarding against software bugs or VLM hallucination.
>
> ### A user study is needed for evaluation
>
> Below, we include a blind user study containing 512 responses and 16 volunteers. The object and method ordering are randomly shuffled in each trial to mitigate bias. Each user is presented with 4 side-by-side videos and asked to rank them (ties is allowed).
>
> | **Method**     | **Avg Rank ↓** | **Best % ↑** |
> |----------------|----------------|--------------|
> | **Pixie**      | **1.7875**     | **55.00%**   |
> | dreamphysics   | 3.7000         | 31.25%       |
> | nerf2physics   | 2.3125         | 28.75%       |
> | omniphysgs     |  2.1625        | 7.50%        |

---

> > ### Author Response · Authors · 2025-11-24
> >
> > ### Why is reconstruction quality included?
> > Our main result and conclusion is based on the realism score provided by the VLM (L107). The reconstruction quality was also computed for reference as reported by some prior works that do test-time reconstruction based on video inputs. The reviewer is correct that our method does not reconstruct any explicit input. However, PSNR and SSIM against the generated labels in PixieVerse can still serve as a coarse signal. Even when there is a spectrum of plausible dynamics, this range is still small compared to the implausible behavior produced by the baselines (e.g., static/stiff or exploding predictions Fig. 5)
> >
> > ### Is it fair to compare baselines on PixieVerse?
> >
> > - It is fair as we evaluate different models by comparing the realism of the resulting videos from their predictions. Pixie model is trained to predict the physics labels which is used by a MPM solver to produce the video. But for evaluation, this process can be considered as a black-box since the gold-standard for comparison is the realism of the end-videos not MSE to the PixieVerse physics labels.
> > - Prior works typically evaluate on a handful of objects like vegetation and deformable objects which are covered by our test-set.
> >
> > ### Quantitative Results on real-world
> >
> > Below we also include the quantitative results on the real-world dataset as suggested.
> >
> > |              | bun   | burger | dog   | Mean  |
> > |--------------|-------|--------|-------|-------|
> > | **PSNR ↑**   |       |        |       |       |
> > | CLIP (ours)         | **25.23** | **23.41** | **20.18** | **22.94** |
> > | RGB                 | 20.31 | 18.92 | 16.37 | 18.53 |
> > | Occupancy           | 19.52 | 18.26 | 15.83 | 17.87 |
> > | OmniPhysGS     | 19.50 | 18.21 | 15.72 | 17.81 |
> > | DreamPhysics   | 20.79 | 19.28 | 16.60 | 18.89 |
> > | NeRF2Physics        | 20.14 | 18.74 | 16.40 | 18.43 |
> > | **SSIM ↑**  |       |        |       |       |
> > | CLIP (ours)         | **0.920** | **0.875** | **0.852** | **0.882** |
> > | RGB                 | 0.866 | 0.821 | 0.797 | 0.828 |
> > | Occupancy           | 0.871 | 0.826 | 0.804 | 0.834 |
> > | OmniPhysGS     | 0.885 | 0.839 | 0.820 | 0.848 |
> > | DreamPhysics  | 0.883 | 0.835 | 0.813 | 0.844 |
> > | NeRF2Physics        | 0.889 | 0.842 | 0.823 | 0.851 |
> >
> >
> >
> > ### Ambiguities in Prediction
> > As we note in the Limitation (L428-430), while there might exists a spectrum plausible parameters, our model would produce a single point estimate from that set. This is still very useful for applications in graphics, computer vision, and robotics. In general, for generation tasks, as opposed to reconstruction tasks, there will always exist some ambiguity. For instance, if we ask Veo3/Sora or text2image model to generate "a picture/video of a cat playing piano", it can still produce some useful outputs in the face of ambiguities (how big is the cat, what breed, what type of piano ect).
> >
> > ### Can we use video model-based optimization like PhysDreamer to label the dataset for training Pixie?
> >
> > This is an interesting future direction! PhysDreamer (L142) firt uses a video generation model to produce a synthetic video of the object in motion. Then, they use test-time optimization to optimize the Young Modulus based on the synthetic video similar to other works in the Related Works (L137-150). However, we'd argue that there are two main challenges with using PhysDreamer for our data annotation currently. For data annotation procedure to be effective, it has to be accurate and preferrably fast. However,
> > - Even SOTA video generation models currently struggle with generate accurate and consistent physics [1]. Relying on such model for annotation can result in poorer quality data. In contrast, while the video generation model in PhysDreamer must generate the entire video taking appearance and physics into account, the job of the VLM in our procedure is much simpler: to generate a set of plausible parameters, which is then simulated by a reliable MPM solver.
> > - Secondly, given the synthetic video, the video-based optimization is still very slow and noisy (Fig. 4), prone to local optima. Due to the highly non-linear optimization process, previous works including PhysDreamer can only estimate one quantity (e.g., Young's Modulus) while requiring the user to hand-tune other paramters (L304-305), defeating the purpose of automatic labeling. Our model and dataset annotation procedure, on the other hand, can inference all physics parameters (including Young's modulus, material class, Poisson's ratio, density) simulatenously (Fig 16).
> >
> > ### L274 sentence
> >
> > This sentence is correct per the the second example in [Merriam-Webster dictionary](https://www.merriam-webster.com/sentences/hence).
> >
> >
> > [1] Physctrl: Generative physics for controllable and physics-grounded video generation, Chen et al 2025

---

> > > ### Comment · Reviewer_dw3d · 2025-11-25
> > >
> > > Thanks for the detailed rebuttal. My initial concerns have been mostly addressed by this rebuttal, and I will raise my score accordingly.
> > >
> > > After the rebuttal, I feel that the strength of this paper outweighs its potential weaknesses. I'm still not that convinced by the scalability of the data annotation pipeline, yet I think it's already a good engineering contribution to the community in the current shape. The explanation of the ambiguity of the problem setting makes sense to me, and I particularly like the analogy to text-to-image models.
> > >
> > > One remaining q: I still don't get how this sentence at L274: "While our method simultaneously recovers all
> > > physical properties, some prior works only predict a subset, hence -." can be grammatically correct. What is the last hyphen after "hence"? In the website you provide, I also don't see such usages.

---

> > > > ### Author Response · Authors · 2025-11-27
> > > >
> > > > Dear Reviewer dw3d,
> > > >
> > > > Thank you for your positive assessment of our work!
> > > >
> > > > In L274, we meant to say that the hyphen refers to the empty entries in the Table (i.e., "N/A" entries) where some methods do not predict those quantities. We will revise the writing to make that more clear.
> > > >
> > > > Please do not hesitate to let us know if there is any more question or issue that we can address.
> > > >
> > > > Thank you.

---

### Author Response · Authors · 2025-11-24

Dear AC and reviewers,

We'd like to thank all reviewers for their suggestions and have incorporated their feedback in the paper. Below we highlighted all new experiments following the reviews:

1. **User study experiment** [dw3d]: our method was consistently ranked higher than the baselines by human participants (Tab. 4).
2. **Real-world quantitative results** [dw3d]: we quantitatively validate Pixie's performance on real-world datasets on the mass estimation on Abo-500 dataset (Tab. 5), and reconstruction quality on Spring-Gauss dataset (Tab. 3).
3. **End-to-end inference time** [wPyp, nTtb]: although geometry and appearance learning is orthogonal to the paper's focus -- physics generation, we still included the preprocessing time. Pixie is still three order magnitude faster than test-time methods.
4. **Pixie as initialization** [74dK]: we also demonstrate Pixie's ability to serve as an informed prior for downstream optimization methods, significantly accelerating convergence (Fig. 19).

---

### Meta-Review · Area_Chair_DCyZ · 2026-01-06

**Summary:**

- Reviewers consistently question whether the PIXIEVERSE dataset and its semi-automatic annotation framework are reliable, scalable, or representative. Concerns include the small number of categories, lack of quantitative validation of annotations, potential bias/noise from VLM-based labeling, and unclear generalization to unseen objects.

- A core concern is that predicting physical parameters purely from static semantic cues is fundamentally ill-posed. Reviewers argue that physical properties are ambiguous, interdependent, and often unobservable from single images or semantics alone, leading to theoretical limitations that scaling cannot fix.

- The evaluation methodology is viewed as vague, incomplete, and sometimes unfair. Reviewers criticize unclear test set descriptions, reliance on VLM-based judgments instead of user studies, missing real-world quantitative comparisons, and comparisons conducted only on the authors’ own dataset.

**Reviewer Concerns:**

- The rebuttal adds evidence of engineering feasibility and partial downstream performance, but it does not provide a direct, dataset-level reliability evaluation or error analysis of the annotation pipeline that reviewers explicitly requested. The ABO-500 mass estimation results partially address one physical attribute but do not validate the full data curation pipeline used in PIXIEVERSE.

- Discussing about ambiguity in generative models (e.g., text-to-image) does not address the specific physics ambiguity raised by reviewers. Unlike visual generation, where multiple outputs can all be acceptable, physical parameters must obey consistency and causal constraints, which the rebuttal does not analyze or formalize.

- Treating the pipeline as a “black box” does not remove the advantage of training on PixieVerse. The Pixie model is explicitly optimized to predict physics labels tailored to this dataset, while baselines were neither trained nor designed with access to these labels or distributions, making the comparison inherently asymmetric regardless of the final metric.

**Reviewer Scores:**

While Reviewer dw3d is positive about this paper, all other reviewers are likely to maintain their negative ratings.

---

### Decision · Program_Chairs · 2026-01-26

Reject